# Top-down control of visual cortex by the frontal eye fields through oscillatory realignment

Domenica Veniero [1✉], Joachim Gross[2], Stephanie Morand[3], Felix Duecker [4], Alexander T. Sack [4] & Gregor Thut[5]

Voluntary allocation of visual attention is controlled by top-down signals generated within the Frontal Eye Fields (FEFs) that can change the excitability of lower-level visual areas. However, the mechanism through which this control is achieved remains elusive. Here, we emulated the generation of an attentional signal using single-pulse transcranial magnetic stimulation to activate the FEFs and tracked its consequences over the visual cortex. First, we documented changes to brain oscillations using electroencephalography and found evidence for a phase reset over occipital sites at beta frequency. We then probed for perceptual consequences of this top-down triggered phase reset and assessed its anatomical specificity. We show that FEF activation leads to cyclic modulation of visual perception and extrastriate but not primary visual cortex excitability, again at beta frequency. We conclude that top-down signals originating in FEF causally shape visual cortex activity and perception through mechanisms of oscillatory realignment.

[1] School of Psychology, University of Nottingham, Nottingham, UK. [2] Institute for Biomagnetism and Biosignalanalysis, University of Münster, Münster, Germany. [3] School of Life Sciences, University of Glasgow, Glasgow, UK. [4] Department of Cognitive Neuroscience, Faculty of Psychology and Neuroscience, Maastricht University, Maastricht, The Netherlands. [5] Centre for Cognitive Neuroimaging, Institute of Neuroscience and Psychology, University of Glasgow, Glasgow, UK. ✉email: domenica.veniero@nottingham.ac.uk

n daily life, we are constantly bombarded by an overwhelming amount of visual information crowded within complex visual scenes. Our limited cognitive resources require a filtering mechanism to prioritize behaviorally relevant input at the expense of irrelevant information. Attention subserves this function. For instance, when the task at hand demands preferential processing of stimuli at a specific location, top-down visuo-spatial attention mechanisms modulate and synchronize neuronal activity within and between neural assemblies in visual areas[1–4] to enable enhanced detectability and discriminability of stimuli that fall within the attended location[5]. Such voluntary visuo-spatial attention mechanisms involve the activation of a complex network of brain areas[6], among which the frontal eye fields (FEF) are key for efficient endogenous attention control[7–9]. Invasive studies in non-human primates have demonstrated that during attentional tasks requiring the voluntary allocation of attentional resources, attention-related signals are initiated in FEF, which subsequently cause activity changes downstream in low-level visual areas[3,10,11]. In addition, non-human primate studies have shown that electrical microstimulation of the FEF causes changes in the activity of visual areas that are comparable to those naturally occurring as a consequence of voluntary deployment of attention[12,13], providing the strongest evidence so far of a causal involvement of FEF in attention control.

In parallel, research on brain oscillations has revealed their role in attention orienting and visual processing[14–16]. The synchronization of specific brain rhythms both within and between brain regions has been proposed to reflect a neural mechanism by which prefrontal areas exert control over visual areas, whereby rhythms in the upper alpha/lower beta-band seem to play a key role in top-down control[15,17–19]. In addition, it has been hypothesized that effects of endogenous attention on perception might rely on the phase reset of ongoing oscillations in primary sensory areas[20–22], which enhances information transfer by aligning periods of high excitability across areas, and at the same time changes the sensitivity of sensory cortex to incoming stimuli[23,24]. However, the causal involvement of brain oscillations in the top-down control of visual cortex excitability remains elusive and the role of phase reset in the visual cortex has been questioned recently[25].

Here, we investigate in the human brain the neural mechanism by which FEF exerts top-down control over visual areas. We apply a single transcranial magnetic stimulation pulse (spTMS) over the right FEF to emulate the generation of an attentional impulse and test if this activation causes the visual cortex excitability to fluctuate as a consequence of a phase reset of brain oscillations generated within the visual areas. In a first experiment, we show that FEF activation by TMS causes an increase in phase consistency in oscillatory beta-band activity over the occipital sites as measured by means of simultaneous TMS-multichannel electroencephalography (EEG). We then examine to what extent FEF activation by TMS also causes systematic changes in visual perception and visual cortex excitability, using a visual motion discrimination task (experiment 2) and TMS-probes of extrastriate (V5) and primary visual cortex (V1) excitability (experiments 3 and 4). In line with the EEG results, we show that as a consequence of FEF activation, motion discrimination and V5 excitability also fluctuate at beta-frequency. For all experiments, we demonstrate that the FEF-induced effects of interest are oscillatory in nature by showing that oscillatory (cosine) models of these responses outperform non-oscillatory (evoked/ERP) models. Taken together, our results reveal a causal, perceptual consequence of FEF-controlled phase-realignment of neural oscillations within visual areas.

## Results

**Experiment 1: Concurrent TMS-EEG**. To test the hypothesis that signals initiated in FEF are changing occipital cortex activity through phase-resetting brain oscillations, we stimulated this prefrontal area with a TMS pulse (to emulate an "attentional" impulse) and traced the effects of this activation in the whole brain through concurrent EEG recordings. Phase reset by the FEF pulse (applied over right FEF) was evaluated in terms of phase consistency across trials (inter-trial phase coherence, ITPC; as in ref. [21]) and compared to a sham-TMS condition, performed to account for nonspecific TMS effects induced by the auditory clicks. EEG responses in the two conditions were statistically compared with a non-parametric cluster-based permutation test including all electrodes and individual time points over a 500 ms time window locked to the FEF-TMS pulse (Active or Sham).

**FEF activation causes phase reset of neural activity in remote occipital areas**. Globally, TMS activation of the right prefrontal area induced an increase in phase consistency across trials lasting up to 300 ms and spanning over several frequency bands (Fig. 1a, see colored boxes for significant time-frequency clusters). Interestingly, these significant differences in phase consistency at distinct frequency bands and latencies were characterized by distinct topographical distributions (Fig. 1a, right maps) suggesting that FEF connects to different areas through specific, frequency-tuned channels. An initial increase in phase consistency peaking at 29 Hz was characterized by a right frontal topography (Fig. 1a: green box/line), with maximal effects around the stimulation site (top map, maximal over FC2). A later increase in phase consistency peaking at 8 Hz (Fig. 1a: blue box/line) was restricted to central and frontal electrodes over the hemisphere opposite to the stimulated area (bottom map, maximal over C3 and FC5). Relevant to our hypothesis, we found a third increase in phase consistency in the low beta-band (Fig. 1a: black box/line) mainly involving occipital channels (middle map, peaking on O2 and Iz; effect magnified in Fig. 1b). As further illustrated in Fig. 1b, the significant difference in occipital phase consistency (comparison of active vs sham FEF-TMS) covered a 200 ms time window and was restricted to a 12–18 Hz frequency band (average effect size $d = 1.94$). The topography of the phase-consistency difference in this band (Fig. 1a: middle map) points to a co-activation of the stimulated frontal and remote occipital sites, with statistically stronger effects over the posterior sites as revealed by t-statistics.

To test whether the TMS pulse is evoking a true oscillatory activity and to exclude that a TMS-evoked potential, i.e., a non-oscillatory signal, is responsible for the changes in low beta-activity, we applied an oscillation-detection analysis (Better Oscillation Detection, BOSC)[26] to the signal recorded from the occipital electrodes (O1, Oz, O2, and Iz) for both the FEF and sham-TMS condition. The BOSC algorithm detects the presence of oscillatory activity by modeling responses from the spectral characteristics of the background activity and identifying segments that deviate significantly from an estimated power and duration threshold of non-oscillatory signals. It thereby rejects any increase in spectral amplitude that is non-repeating over time (see "Methods" section for details). The results indicate the presence of true oscillatory episodes at frequencies between 12 and 20 Hz over each posterior electrode of interest in the 300 ms following FEF activation, as compared to the sham control (see Supplementary Fig. 1).

Taken together, these results suggest that a brief activation of FEF by TMS caused a selective phase alignment in the beta-band at electrodes over occipital sites.

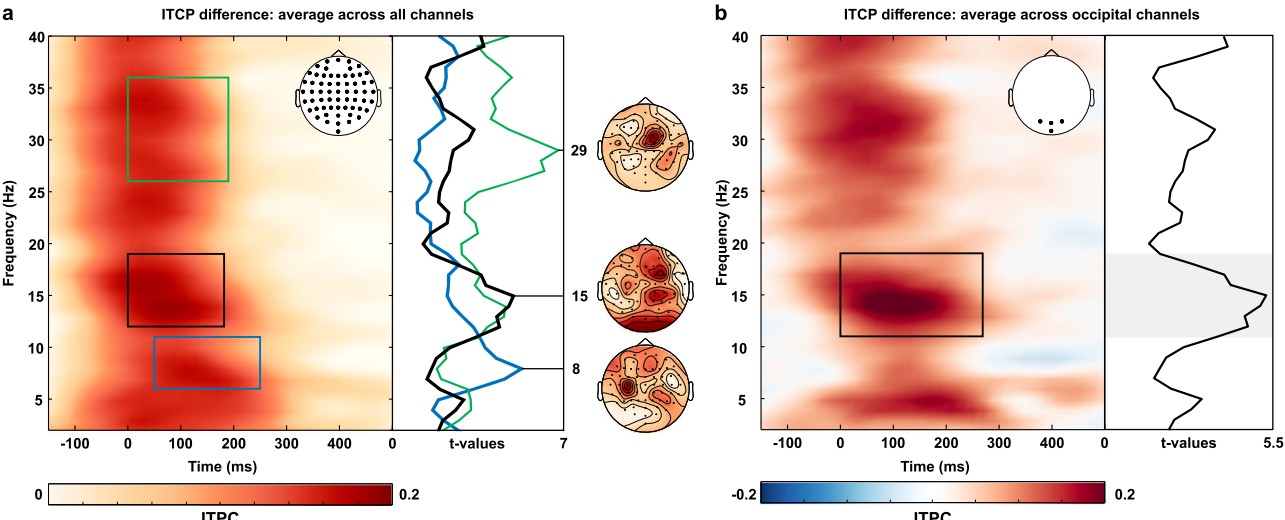

**Fig. 1 Effects of FEF activation on EEG activity as measured by inter-trial phase coherence (ITPC).** ITPC across the whole scalp (**a**) and over occipital electrodes (**b**) is increased after FEF-TMS (relative to sham TMS; $n = 11$ participants). **a** Time-frequency plot of ITCP difference between FEF- and sham TMS indicates frequency- and site-specificity of FEF-activation effects. The *t*-values in the right panel show averages across electrodes per significant time-frequency cluster, marked by the rectangle of the same color in the left panel. ITPC increased around 29 Hz over the right frontal sites (next to the stimulation target) from 0 to 200 ms after TMS pulse delivery, around 15 Hz over posterior electrodes from 0 to 200 ms after the TMS pulse; and around 8 Hz over central and frontal electrodes contralateral to the stimulation target from 30 to 250 ms post-stimulus. Maps on the right indicate the topographical distribution of *t*-values for the frequency bands highlighted by each rectangle. **b** Average effect over occipital sensors. Significant differences between conditions (FEF- vs. Sham TMS) are highlighted by the rectangle and indicate a phase reset between 12 and 18 Hz. The plot on the right depicts the *t*-values as a function of frequency (2–40 Hz) over the significant time window (0–200 ms).

**Experiment 2: Behavioral effects of FEF-spTMS.** Next, we sought to test whether the rhythmic fluctuation of occipital cortex activity initiated by the FEF pulse is perceptually relevant. If this were the case, perception of visually presented stimuli should exhibit a rhythmic fluctuation, time-locked to underlying oscillatory activity and hence to the FEF pulse. As for the EEG experiment, we emulated the attentional signal by spTMS over the right FEF but then tested the top-down effects on visual processing with a motion discrimination task. Participants were asked to judge the direction of moving dots presented centrally (Fig. 2). The motion stimuli were of 35 ms duration, lasting long enough that motion direction is perceived (mean accuracy = 70 ± 12%, significantly > chance, $t_{(10)} = 5.4$, $p < 0.01$; see ref. [27] for similar motion durations), but being short enough (fraction of one beta cycle) to be able to test phase-specificity of perception in the beta-band. We reasoned that if the phase alignment of oscillatory activity in the occipital cortex by the attentional impulse is perceptually relevant, we should be able to reveal the induced periodicity also in perception by sampling motion discrimination accuracy at various time points across trials. The motion stimulus was presented at 24 different delays after FEF activation, covering a time window of 270.6 ms, in 11.8 ms steps (Fig. 2). This design yielded a time course of visual performance (discrimination accuracy) post FEF activation that was analyzed for the presence of cyclic patterns by fitting cosine models between 7 and 25 Hz to the data (see "Methods" section for a detailed description).

**FEF activation causes periodic fluctuation in the perception of motion stimuli.** To take into account the transient nature of phase reset (see TMS-EEG data), behavioral data were split into 2 partially overlapping time windows, covering 200 ms each. The partial overlap was necessary to include enough data points for a sufficient frequency resolution in the tested bands (7–25 Hz). Figure 3a shows the time course of TMS-locked motion discrimination accuracy per window. The emulated attentional

impulse over FEF caused visual performance to fluctuate over time over both time windows and, in line with the EEG results, the frequency best explaining the behavioral performance pointed to an underlying low beta-frequency (window1: 17 Hz; window2: 19 Hz; see the red line of best fit in Fig. 3a). To test for statistical significance, cosine models were fitted to each participant's data, yielding R-squared values for each frequency in the 7–25 Hz range, which were then subjected to a bootstrap procedure. As shown in Fig. 3b (middle panels), fluctuations in discrimination accuracy were significantly explained by cosine models in the beta-band in both time windows, ranging from 15 to 23 Hz in window1 (9 significant frequency bins) and from 17 to 24 Hz in window2 (8 significant frequency bins). For the second window only, cosine models from 7 to 10 Hz were also significant. We then tested the single-participants' model fits to behavioral data for the presence of phase consistency across participants (using Rayleigh tests for non-uniformity of circular data), considering the cosine models that significantly fitted the data. We found significant phase consistency across participants for the lower range of the beta-frequency bins again in both time windows, i.e., 14–17 Hz in window1; and 17–19 Hz in window2 (see polar plots in Fig. 3b and Supplementary Fig. 2, polar plots illustrate individual phase data (=blue dots) being clustered around the average phase (=red dot)). No other frequency showed phase consistency. The bias for lower beta frequencies is likely due to faster frequencies suffering more from phase jitter and hence from averaging out across participants.

Finally, as for experiment 1, we tested whether the fluctuation of visual performance in the beta-band could be explained by an alternative, non-oscillatory model. Our aim was to rule out that changes in behavioral performance could be ascribed to an ERP-like response generated by FEF activation. To this end, we fitted a non-oscillatory model to the behavioral data for comparison to the oscillatory (cosine) model. The ERP-like response was modeled as an exponentially decaying sinusoid (as in ref. [28]) using frequencies between 7 and 25 Hz (see "Methods" section for

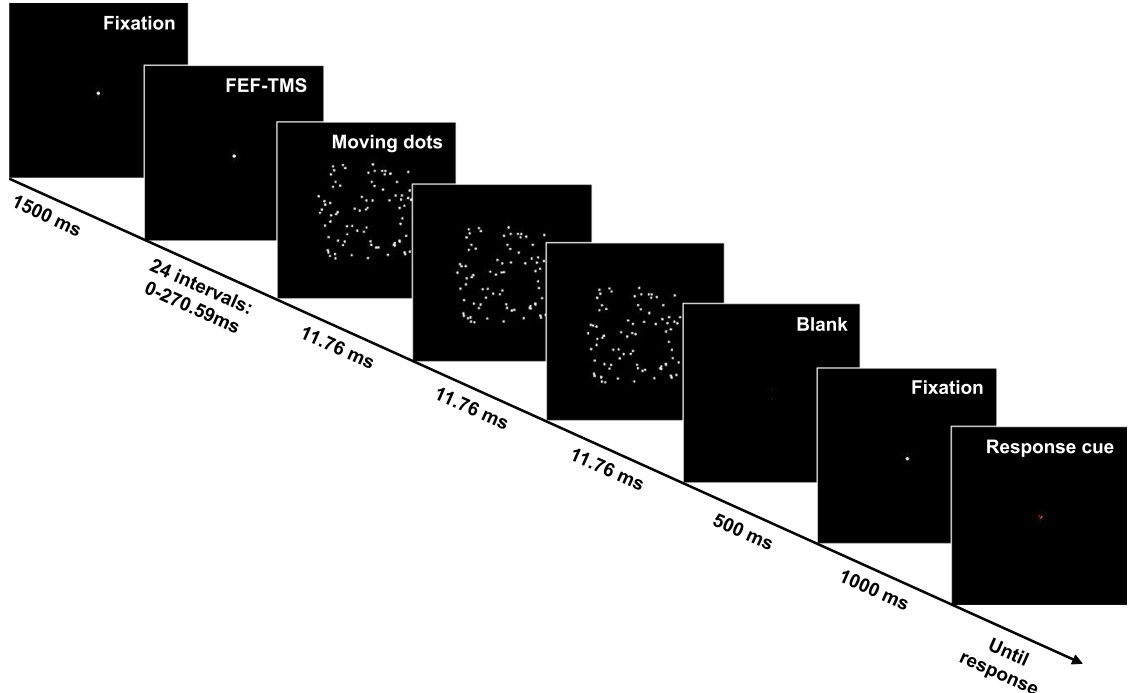

**Fig. 2 Schematic representation of the motion discrimination task.** A white fixation dot was presented for 1500 ms. A TMS pulse was then applied over the right FEF and, after a variable delay ranging from 0 to 270.6 ms, moving dots were presented for 3 frames. After a blank of 500 ms, the fixation dot reappeared for 1000 ms. Participants were instructed to report perceived motion direction when the fixation dot turned red. For illustrative purposes, the size of the visual stimuli has been magnified.

a detailed description). To confirm that the cosine model best explained the behavioral data, we compared the two models using Akaike's Information Criterion (AIC) for each window (Fig. 3c). We found that the cosine model consistently yielded a lower AIC score as shown by individual data points in Fig. 3c (window1: ERP mean AIC = 109.66, sd = 6.75, Cosine AIC mean = 99.62, sd = 7.81; window2: ERP AIC mean = 108.12, sd = 6.25, Cosine AIC mean = 96.47, sd = 8.55) indicating that the cosine model was the qualitatively better model. The average difference in AIC scores (AIC ERP minus AIC cosine) was $\Delta = 10$ for window1 and $\Delta = 11.6$ for window2, confirming that the oscillatory model better explained the behavioral data. Note that a difference in AIC of 10 between models is considered strong evidence in favor of the model with lower AIC score[29].

In summary, these results complement the EEG findings showing that emulated, attentional top-down control signals by brief FEF stimulation do not only phase-reset oscillatory activity in the beta-band but also caused visual performance to cycle at the same frequency.

**Experiments 3 and 4: Dual-site TMS.** Based on the results of experiments 1 and 2, we designed two follow-up experiments to further investigate the causal link between FEF-generated signals and the phase alignment of oscillatory activity within visual areas. Experiments 3 and 4 were designed to answer three questions.

First, we wanted to establish whether the cycling of perception with oscillatory phase alignment may be explained by a modulation of visual cortex excitability. To this end, we used a dual-site, double-pulse TMS protocol. As in the first two experiments, we emulated the attentional FEF signal by spTMS over the right FEF but then tested the downstream effects in the visual cortex directly with a second TMS pulse over the occipital cortex at different delays from the FEF pulse. The second pulse, hereafter referred to as visual test pulse, served to measure the

excitability of the visual areas by evoking phosphenes, which are illusory visual percepts known to be generated in visual areas. In line with the results of the first two experiments, we expected the FEF activation by TMS to induce a periodicity in visual cortex excitability as revealed by samples of phosphene perception at various time points post FEF activation.

Second, taking advantage of the spatial resolution of TMS relative to EEG, we wanted to test whether FEF-spTMS can equally or differentially influence the excitability of extrastriate (V5) versus early visual cortex (V1). To this end, we tested two independent groups of participants. In experiment 3, right FEF activation was followed by a visual test pulse over right V5 evoking moving phosphenes (FEF-V5 TMS). In experiment 4, right FEF activation was followed by a visual test pulse over right V1 evoking static phosphenes (FEF-V1 TMS). In each experiment, participants were instructed to report the presence of phosphenes after the delivery of the visual test pulse, which was applied at one of 19 possible time points after FEF activation (in 15 ms steps) covering a time window of 300 ms (slightly longer than the window of significant phase consistency in experiment 1).

Third, we wanted to confirm that our findings were the results of neural interactions between FEF and occipital areas and could not be ascribed to TMS-unspecific effects. To this aim, for each experiment, we run a dual-site TMS control condition, where the attentional pulse was replaced by an active TMS control (spTMS over the vertex). Vertex TMS followed by right V5 TMS (Cz-V5 TMS) served as a control for experiment 3 and vertex TMS followed by right V1 TMS (Cz-V1 TMS) as a control for experiment 4.

This design yielded a time course of phosphene perception rate (known to index visual cortex excitability) post FEF activation for each experiment (FEF-V5, FEF-V1) that was controlled for unspecific TMS effects (partialling out Cz-V5 and Cz-V1 effects, respectively) and then analyzed for the presence of

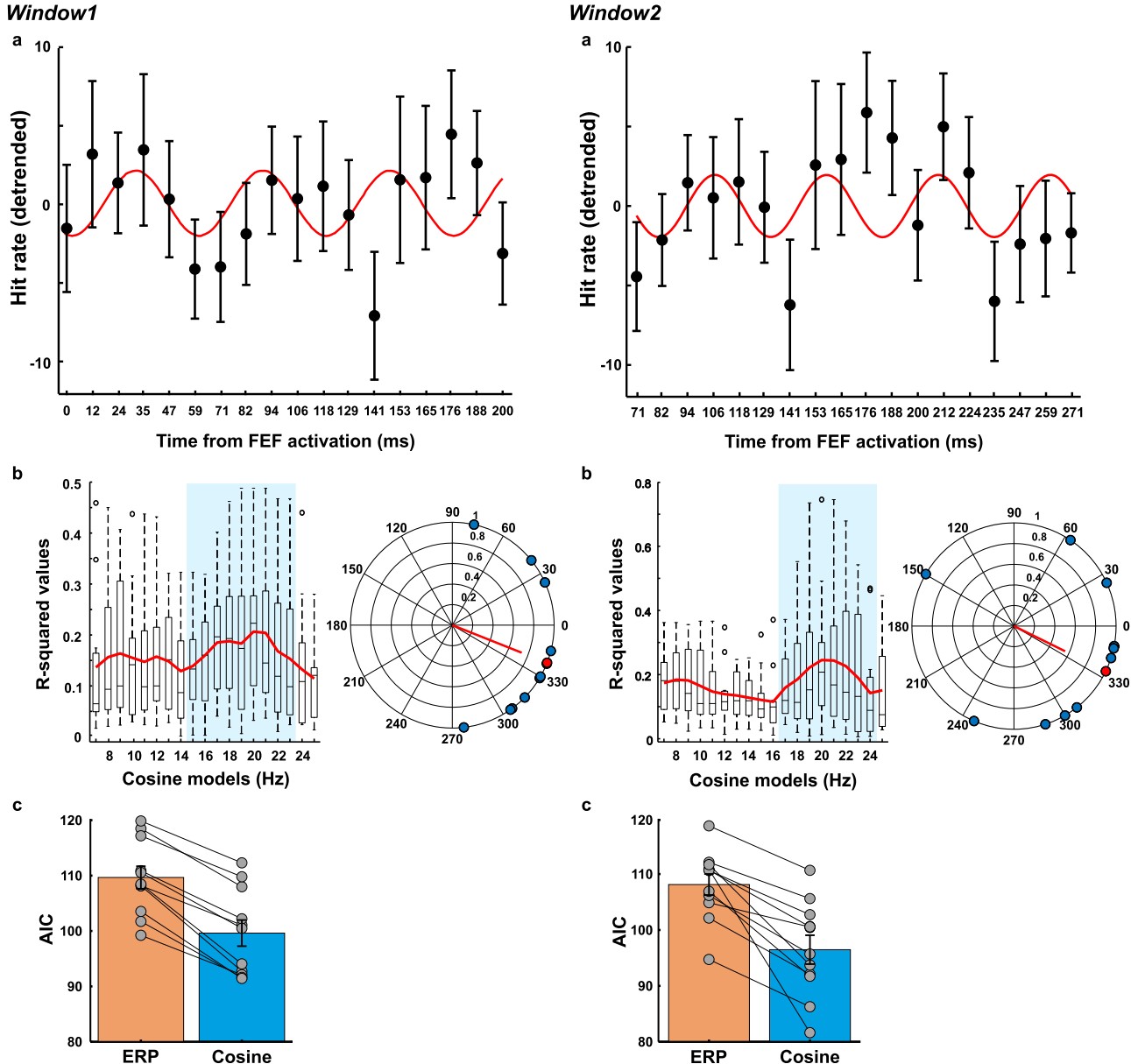

**Fig. 3 Effects of FEF activation on perception over time as measured by participants' performance in a motion discrimination task (discrimination accuracy).** Single-pulse TMS was used to activate the right FEF and moving stimuli were presented at different intervals (0–271 ms after FEF activation) to measure motion perception ($n = 11$ participants). **a** Average performance (linearly detrended, ±sem) for each time point for window1 (0–200 ms) and window2 (71–271 ms), respectively. The red line indicates the best-fitting cosine model on averaged data (window1: 17 Hz and window2: 19 Hz). **b** R-squared-values as a function of fitted frequency per window (averages from individual fits in red), with cosine models significantly fitting the data and showing significant phase consistency highlighted in blue. Median ± whiskers with maximum 1.5 interquartile range (IQR) and outliers are shown for each cosine model. The polar plots show the phase distribution across participants (in blue) and the average phase (in red) for the best-fitting model at group level, for which phase of the best-fitting cosine function was significantly different from a uniform distribution. See Supplementary Fig. 2 for the corresponding phase information extracted from the EEG signal for comparison to the behavioral phase data (data shown for first window only). **c** Akaike's Information Criterion (AIC) for the 2 models fitted to the data. Blue bar: oscillatory (cosine) model. Yellow bar: non-oscillatory (ERP) model. Gray dots represent AIC scores per model for each participant ($n = 11$), whereby the model with the lowest AIC value among all possible frequencies in the significant range (highlighted in blue in **b**) was selected for each participant (error bars: +sem). For window1 the average difference between ERP AIC and cosine AIC was $\Delta = 10$; for window2 $\Delta = 11.6$, thus providing strong evidence in favor of the oscillatory model better explaining the behavioral data[29]. Source data are provided as a Source data file.

cyclic patterns by fitting cosine models between 7 and 25 Hz to the data. As for the motion discrimination task, moving phosphene data were analyzed by statistically evaluating R-squared values obtained from cosine model fits per participant and condition, and the data were split into two overlapping windows covering 200 ms.

**FEF activation causes periodic fluctuations in V5 excitability.** Figure 4 shows the time course of moving phosphene perception rate as tested by V5 stimulation over the first 200-ms window following FEF-spTMS (Fig. 4a) or control (vertex) spTMS (Fig. 4b). The emulated attentional impulse over FEF caused the moving phosphene perception rate to fluctuate over time in a

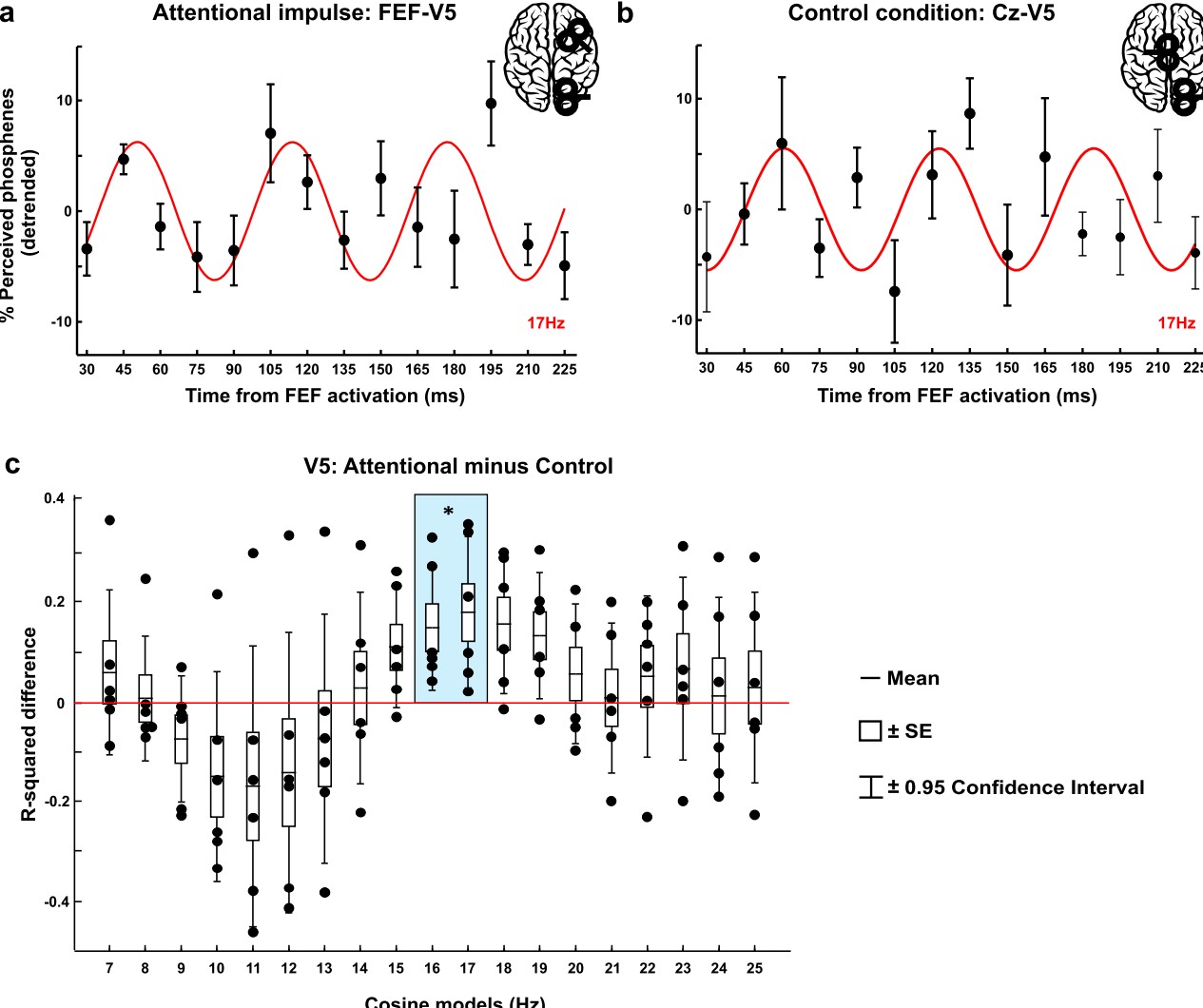

**Fig. 4 Effects of FEF activation on V5-excitabilty over time as measured by moving phosphene perception rate.** A double TMS coil design was used to place an initial FEF-TMS pulse (**a**) or a control (vertex) TMS pulse (**b**), followed by a second, V5-TMS test pulse evoking moving phosphenes at different delays (identical for **a** and **b**) in $n = 6$ participants (within-groups design). Each panel depicts the average percentage of induced phosphenes (linearly detrended, ±sem) for each time point. The red line indicates the cosine model (17 Hz) for which the two conditions differ, and that significantly explains phosphene perception fluctuations for FEF-V5 (**a**) but not Cz-V5 (**b**). **c** R-square differences between FEF activation and control (vertex) condition for each fitted cosine model from 7 to 25 Hz. To test for significant differences, we used a permutation test and considered data significant if they fell above the 97.5th percentile of the null distribution. Significant differences in the beta-band were confirmed by a repeated measure ANOVA (Condition × Frequency interaction; $F_{(18,90)} = 2.737$, $p = 0.001$, $\eta p2 = 0.35$) and follow-up two-tailed $t$-tests (16 Hz: $t_{(5)} = 3.04$, $p = 0.029$; $d = 1.24$; 17 Hz: $t_{(5)} = 3.07$, $p = 0.028$; $d = 1.25$). The bounds of the boxplot indicate ±sem, the whiskers represent ±0.95 confidence interval, the horizontal black line the mean of R-squared values. The blue rectangle highlights the significant differences between conditions as indicated by a non-parametric permutation test. Black dots represent single-subject data. Source data are provided as a Source data file.

more consistent beta pattern than control (vertex) stimulation, as suggested by Fig. 4 (compare sem in panel a vs. panel b in relation to the best beta cosine fit = red line). A comparison between these two conditions as to the frequency content in the phosphene curve over time confirmed the presence of beta oscillations in moving phosphene perception after FEF-spTMS, relative to Control TMS (Fig. 4c: window1 (samples from 30 to 225 ms)). The 16–17 Hz model significantly deviates from zero-difference between the main and control condition (see Fig. 4c, deviation from zero line), as revealed by a permutation test on mean differences for each cosine model. These results were further supported by a repeated measure ANOVA with factors Condition (FEF-V5 vs Cz-V5) and Frequency (7–25 Hz in 1-Hz-step), indicating a significant Condition × Frequency interaction ($F_{(18,90)} = 2.737$, $p = 0.001$, $\eta p2 = 0.35$). Follow-up tests (uncorrected for tests across

multiple frequencies) confirmed that FEF-V5 and Cz-V5 were significantly different for the 16 Hz ($t_{(5)} = 3.04$, $p = 0.029$; $d = 1.24$), 17 Hz ($t_{(5)} = 3.07$, $p = 0.028$; $d = 1.25$) and 18 Hz cosine model ($t_{(5)} = 2.86$, $p = 0.035$; $d = 1.17$). This corroborates the presence of beta-cycles in V5 excitability and shows that this cannot be explained by unspecific TMS effects, such as the sound or somatosensation associated with TMS delivery, but are due to FEF activation, and hence to top-down interactions from FEF to visual areas.

After having established specificity of our results to FEF stimulation, we examined these fluctuations in more detail within our main (FEF-V5 TMS) condition. As for the motion discrimination task, we tested for the presence of specific frequencies in phosphene perception fluctuations over time based on goodness of fit (R-square values) using a bootstrap procedure, and whether there was phase consistency/reset (defined here as

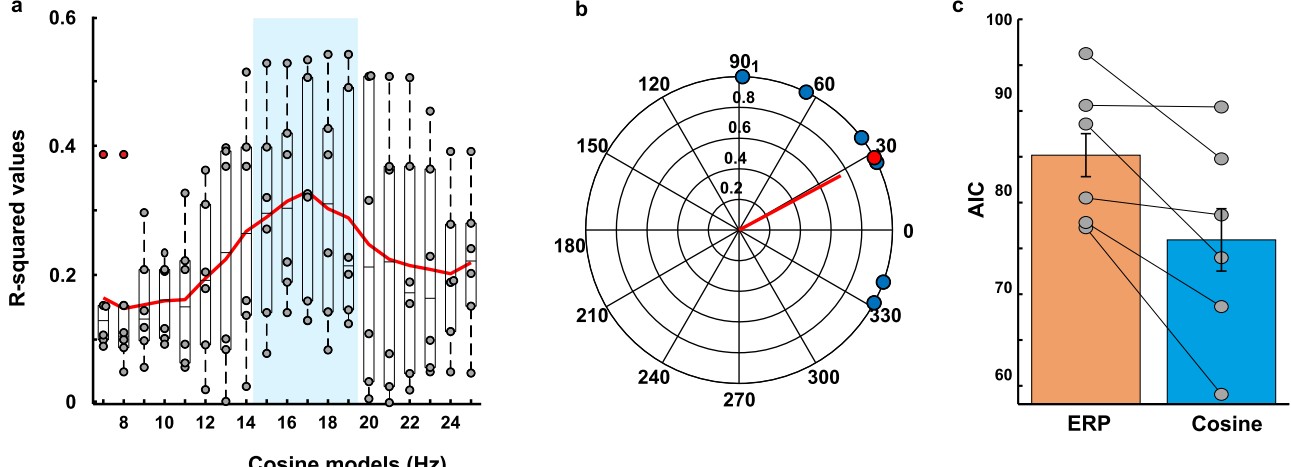

**Fig. 5 Rhythmicity in V5 excitability probed with moving phosphene perception after a FEF-TMS pulse. a** R-squared-values are reported for window1 (red line: averages from individual fits; dots: single-subject data, $n = 6$ participants), with cosine models significantly fitting the data highlighted in blue. Median ± whiskers with maximum 1.5 interquartile range (IQR) and outliers (red dots) shown for each cosine model. Gray dots represent individual data. **b** Polar plot indicates the phase distribution across participants (in blue) and the average phase (in red) at 17 Hz, for which phase of the best-fitting cosine function was significantly different from a uniform distribution. **c** AIC values for each model (cosine, ERP) fitted to the data. Gray dots represent individual values. The model with the lowest AIC value among all possible frequencies in the significant range (highlighted in blue in **a**) was selected for each participant for average and statistics (error bars: +sem). The average difference between ERP AIC and cosine AIC was $\Delta = 9.2$, thus providing strong evidence in favor of the oscillatory model better explaining the behavioral data[29]. Source data are provided as a Source data file.

non-random phase distribution) across participants. This revealed that cosine models from 15 to 19 Hz significantly explained moving phosphene fluctuations, while no other frequency fitted this data (effects were beta-specific) (Fig. 5a). When testing the FEF-triggered moving phosphene curve for phase-locking—calculated as phase consistency across participants—we found significant results in the beta-band (16–17 Hz) ($p < 0.01$) (Fig. 5b polar plot illustrates individual phase data (=blue dots) being clustered around the average phase (=red dot)).

Finally, as for experiment 2, we evaluated whether an alternative, non-oscillatory (ERP) model could better explain these (FEF-V5) data (see Fig. 5c). We calculated and compared the AIC values of the cosine and the ERP models. The results directly confirmed that the cosine model better explained the phosphene fluctuation over time (ERP AIC mean = 85.17; sd = 7.78, Cosine AIC mean = 75.93; sd = 11.29). Given the average difference between ERP and cosine AIC of $\Delta = 9.2$, we conclude that the ERP did not represent a plausible model in comparison to the oscillatory model[29].

When considering the second 200-ms window, covering data samples from 105 to 300 ms, we found no significant difference in moving phosphene perception between conditions (FEF-V5 vs Cz-V5), nor any evidence for a significant phase reset induced by brief FEF activation, suggesting that the effects on V5 excitability were restricted to the early 200-ms window (see Supplementary Fig. 3a, b).

In summary, these results complement the EEG and behavioral findings (experiments 1 and 2) showing that FEF activation causally shapes V5 excitability through a phase reset of beta-activity. As we failed to find any significant result in the second time window when testing against an active TMS control (experiment 3), and in line with the timing of the FEF-TMS triggered periodic EEG activity over occipital cortex (~200 ms window for occipital cluster, Fig. 1b), we conclude that the top-down influences from FEF on oscillatory activity in visual areas last up to 200 ms (~3.4 cycles for 17 Hz beta-activity; see also Supplementary Fig. 1).

**FEF activation fails to modulate V1 excitability**. Figure 6 shows the results for FEF influences on V1 excitability. Statistical

analysis performed on the R-squared values of cosine model fits to the FEF (FEF-V1) and control (Cz-V1) data revealed no difference for any frequency, neither when testing the difference with permutation tests nor with a repeated measure ANOVA (Condition × Frequency: $F_{(18,108)} = 0.24$, $p = 0.999$). This indicates that the attentional impulse over FEF did not influence primary visual cortex excitability differentially as compared to Cz-TMS. The same pattern also emerged when considering the second time window (see Supplementary Fig. 3c, d). Given the absence of any difference between the main condition and its control, no further analysis was carried out.

## Discussion

The aim of this study was to test the neural mechanisms by which FEF exerts top-down control over visual areas. In analogy with animal studies investigating the causal influence of prefrontal areas on the visual cortex through microstimulation, we activated the right FEF by means of a brief non-invasive brain stimulation pulse and concurrently recorded EEG activity from the whole brain. Our results revealed an instantaneous, FEF-TMS triggered phase reorganization of EEG beta-activity over remote occipital sites. To investigate whether the phase reset of neural beta-activity is perceptually relevant (and therefore has a causal role in shaping perception), we conducted a series of follow-up experiments. These revealed that the brief FEF activation also caused a cyclic modulation of visual perception and visual cortex excitability, as evidenced by the induced fluctuations of motion discrimination accuracy and perceived moving phosphenes, both at beta-frequency. We also provided evidence that FEF activation influenced V5- but not V1 excitability. It is worth mentioning that the follow-up experiments performed to collect behavioral data (visual performance and excitability measures) do support and enhance the EEG findings in several ways. They compensate for the coarse spatial resolution of EEG by revealing anatomical specificity of the periodic fluctuations to extrastriate (V5) excitability. They corroborate phase reset in the beta-band as a mechanistic account of the top-down effects, given the significant phase consistency of perceptual beta-cycles across participants, and firmly link the occipital EEG oscillation to visual function.

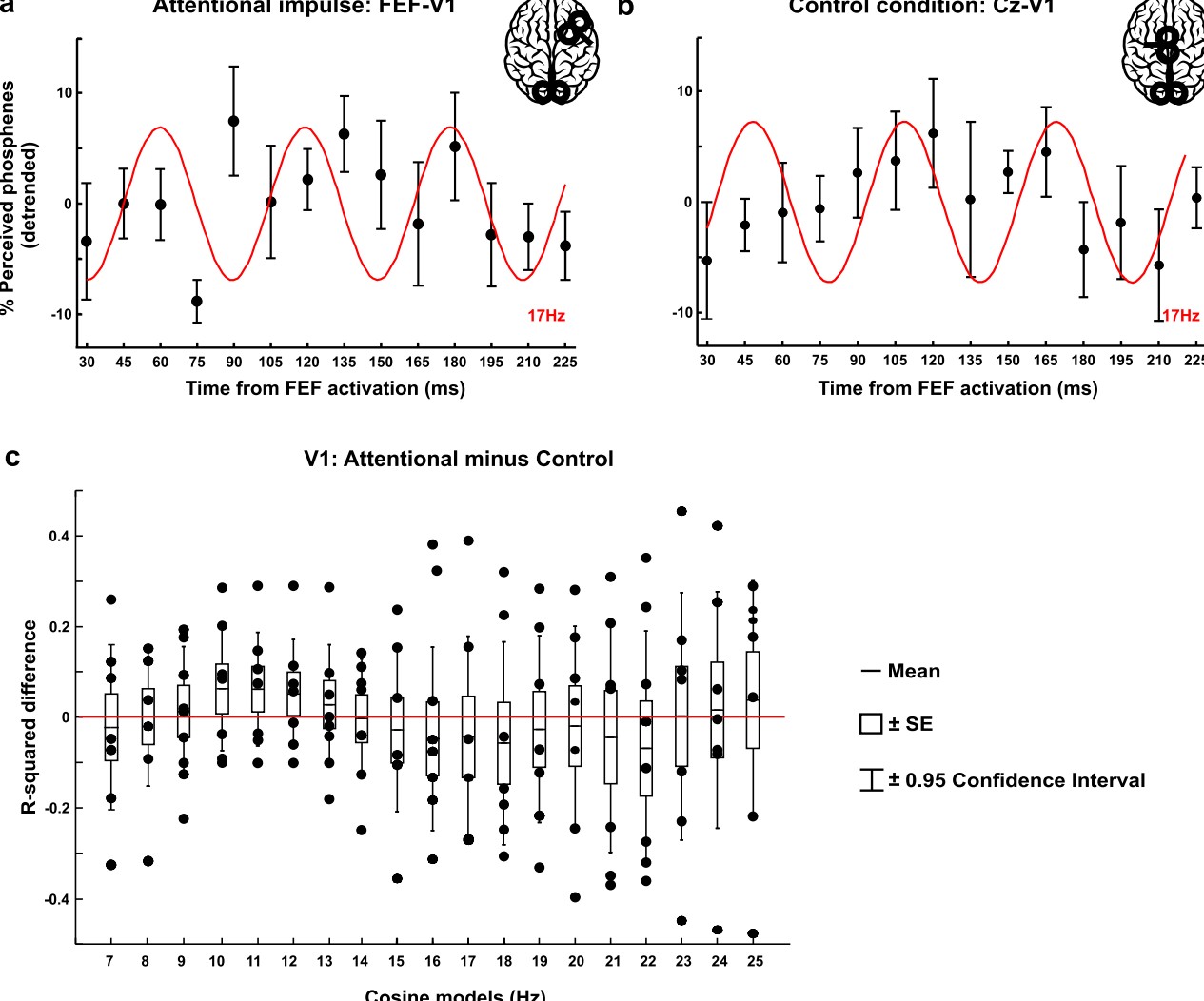

**Fig. 6 Effects of FEF activation on V1-excitabilty over time as measured by phosphene perception rate.** The design was identical as in Fig. 4, but the TMS test pulse was delivered to V1 instead of V5 ($n = 7$ participants, within-groups design). The average percentage of phosphenes evoked by V1 stimulation (linearly detrended, ±sem) at different time points for the FEF (**a**) and control (**b**) condition, represented together with the best-fitting cosine functions (red line) at 17 Hz (not significant) for comparison with Fig. 4a, b. **c** R-squared differences between attentional and control condition for each fitted cosine model. To test for significant differences, we used a permutation test and considered data significant if they fell above the 97.5th percentile of the null distribution. We found no difference between conditions. For each cosine model, the bounds of the boxplot indicate ±sem, the whiskers represent ±0.95 confidence interval, and the horizontal black line the mean R-squared values. Black dots represent single-subject data. Statistical analysis did not reveal any significant difference between conditions. Source data are provided as a Source data file.

Finally, they show that the effects are specific to FEF stimulation as these effects were not observed when FEF-TMS was replaced by an active (vertex) TMS control, accounting for a number of alternative explanations (e.g., auditory and somatosensory co-stimulation[30,31]). Collectively, the results of our four experiments hence provide converging empirical support for top-down control from FEF on visual cortex activity and function to be mediated by phase realignment of neural oscillations at beta-frequency.

The phase of ongoing oscillations reflects the excitatory state of neural populations at any given moment and therefore influences the outcome of sensory processing[32,33] with stimuli arriving at high excitability phases being processed more effectively[34,35]. Therefore, it seems plausible that attentional impulses will prioritize task-relevant stimuli by reorganizing the phase of neural oscillations generated within sensory areas, such as the visual cortex[32,36]. There is evidence that attention can bias the phase of relevant oscillations to suppress the processing of irrelevant information[37] or optimize target discriminability[38] (but see

ref. [25]), if a cue allows for temporal predictions about the forthcoming stimulus (see also studies on speech processing, e.g., ref. [39], and rhythmic predictions, e.g., refs. [20,21,40]). However, direct tests of such top-down control over oscillatory phase in sensory areas are still missing[41]. Our finding of a brief FEF activation causing an instantaneous, perceptually relevant phase reorganization of both neural activity and excitability within the visual cortex (V5) reveals that oscillatory phase realignment is a fundamental building block of top-down control, because showing that this effect is observable when the relevant area of the attention network is directly stimulated, in addition to experimental manipulation of temporal predictability through cueing or external rhythms as shown previously.

The top-down effects were frequency-specific since the phase reset was restricted to the beta-band in both the neural and perceptual measures. This result is in line with previous findings showing that top-down processing is predominantly associated with beta oscillations both within sensory areas[42–44] and within

large-scale fronto-occipital networks ([4,10,15,17–19,34,45], for a review, see ref. [46]). In our experiments, we did not find any evidence for alpha-band modulation in the 8–12 Hz frequency band despite its prominence in sensory and attention functions, e.g., refs. [14–16]. This is in contrast with previous TMS studies investigating the effect of FEF stimulation on brain oscillations. When applied over FEF, inhibitory offline or online rTMS protocols (such as cTBS, low frequency rTMS, or short burst rTMS as applied in refs. [47–49]) have been reported to interfere with the allocation of covert attention and the known EEG signatures of spatial attention orienting in the alpha-band. While the choice to limit the analysis to the alpha-band in these studies prevents a direct comparison with our results, differences in the experimental design, including the need for sustained attention deployment in previous but not our study, might explain some of the differences in the results. Moreover, the lack of alpha modulation in our study may be explained by previous findings indicating that while beta-activity could be related to processing of relevant stimuli (i.e., target)[4,22], alpha oscillations might be involved when the task at hand requires the inhibition of competing information (distracters or locations)[50,51], which was not the case in our experiments. Since we observed beta-band phase reset in EEG data in a resting condition (experiment 1), for performance during a visual task (experiment 2) and visual excitability in the absence of visual stimuli (experiments 3 and 4), we conclude that reorganization of beta phase may represent a basic code for top-down (FEF-to-visual cortex) communication. However, this does not rule out that oscillatory activity under top-down control might change with task demands (see also next paragraph), the probing of other higher-order attention areas than FEF or that cross-frequency effects not examined here may also be at play[52].

Several behavioral studies have already revealed frequency-specific fluctuations in performance measures to become apparent immediately following the presentation of a discrete, sudden-onset sensory event[53–56] (for a review see ref. [33]). Some of these studies have also shown that this rhythmicity in behavior matches the periodicity of concurrently recorded EEG[55,56], or follow-up, offline MEG[57,58]. These fluctuations in performance are likely due to a reset of attention to the position and/or timestamp of the external event, followed by periodic exploration of the visual scene[54]. Here, we capitalized on these designs, but unlike the above studies, used TMS to emulate the "attention" signal by directly activating a node of the attention system (FEF), rather than through presenting a sudden-onset sensory event capturing attention. In contrast to our findings, these previous studies have provided evidence of an attentional sampling at slower frequencies in the theta- to alpha-bands[53,54]. This discrepancy can be ascribed to differences in task demand, given that our design did not favor one location over the other (i.e., there was no attentional cue to direct attention to one visual field as in ref. [53]) and did not encourage to sample multiple locations (as in ref. [54]) that will have caused sustained spatial (re)orienting. Furthermore, our design probed a specific attention circuit (FEF-visual areas) and its mode of top-down communication, while the studies above used visual cues for phase reset that will have likely engaged additional circuits of the attention system.

Following the activation of the right FEF, we found the top-down effects to be anatomically specific, i.e., limited to V5. The lack of effects on V1 supports previous evidence from electrophysiological and imaging studies of a downstream progression of attentional effects[59] revealing that feedback signals are first sent to higher-order visual areas, such as V5, before being retransmitted to earlier areas, such as V1, generating a gradient in the magnitude of attentional effects, which are maximal over extrastriate

and limited over primary visual area (for human studies see ref. [60]; for animal studies see ref. [61]). It is worth noting that Ruff and colleagues[62] reported a change in BOLD signal within V1 when FEF was stimulated but using short bursts of TMS pulses. It is conceivable that the single TMS pulses we used were too weak to reach V1 and therefore only able to activate direct projections to extrastriate areas.

In conclusion, by showing that the FEF-triggered periodic fluctuation in discrimination accuracy and phosphenes perception over V5 cycles at beta-frequency and coincides with the alignment of occipital beta phase as recorded with EEG, we provide direct evidence that top-down signals can change the phase of relevant oscillatory activity in a perceptually relevant manner.

## Methods

**Participants**. A total of 26 right-handed volunteers were recruited. Twelve took part in the first two experiments (experiments 1 and 2; 3 men; mean age = 26, sd = 3.9), but one participant was excluded due to an unstable performance in the motion discrimination task. For experiments 3 and 4, a total of 14 participants were enrolled. After an initial phosphene training session (see experimental procedure for details), which tested the ability to reliably report moving or static phosphenes when single-pulse TMS was applied to V5 or V1 respectively, half of the volunteers (n = 7 participants) were assigned to the V5 group/experiment 3 (all females; mean age = 26, sd = 4.1) and the other half (n = 7 participants) to the V1 group/experiment 4 (all females; mean age = 23; sd = 2.5). Three participants were recruited for both experiments 3 and 4. One participant was excluded from the first (V5) group because phosphenes were not reliably reported during the second session.

All participants had normal or corrected-to-normal vision and reported no contraindication to TMS or any neurological, psychiatric, or relevant medical condition. All protocols were performed in accordance with ethical TMS standards and approved by the Ethical Committee of the College of Science and Engineering (University of Glasgow). Written informed consent was obtained prior to each experimental session.

## Procedure

*Experiments 1 and 2: TMS-EEG and motion discrimination*. Participants were seated in a comfortable armchair in a dimly illuminated room, with their chin on a chinrest to ensure a stable head position, placed 57 cm from the screen. The experiments consisted of three conditions: While EEG was continuously recorded, TMS was applied in two TMS-only conditions (active and sham TMS) or combined with a two-alternative forced-choice motion discrimination task (TMS-task).

For the TMS-only conditions, an active or sham-TMS pulse was applied over right FEF 1500 ms after the appearance of a fixation dot on the screen. Following TMS, the fixation dot was kept on the screen for an additional 1500 ms, after which the fixation dot turned red, which prompted the participants to press a key to move to the next trial. A total of 100 active TMS and 100 sham-TMS pulses were applied over the right FEF.

For the TMS-task condition, right FEF TMS was followed by a central motion stimulus (see Fig. 2). Stimuli were presented on a CRT monitor (85 Hz refresh rate, 1280 × 1024 pixel resolution) using E-prime 2.0 (Psychology Software Tools, Pittsburgh, PA, USA). Each trial began with a white fixation dot (3 × 3 pixels). In the TMS-task condition, the fixation dot was followed after 1500 ms by a square patch (3° × 3° visual angle) of 80 white dots (3 × 3 pixels) presented on a black background at the center of the screen. A percentage of dots moved either rightward or leftward (coherent motion) over three frames (35 ms duration) at 4.4°/s, whereas the remaining percentage of dots moved in a random manner. Pilot data confirmed that participants perceived motion with the 35-ms-coherent motion stimuli (motion discrimination accuracy above chance). After a 500 ms blank, the fixation dot reappeared for 1000 ms. A change in the fixation dot color from white to red prompted the participant to indicate the motion direction of the stimulus, by pressing either a left or right response key with the index or middle finger of the right hand. The next trial started as soon as the response was made. The coherent motion stimuli were preceded by a single TMS pulse over right FEF. To study cyclic fluctuation in visual perception, the interval between FEF activation and the presentation of the visual stimuli was randomly chosen in each trial from 24 possible delays (0–270.6 ms in steps of 11.8; i.e., 1 screen frame) (Fig. 2). Each interval was tested 14 times (7 for each motion direction), for a total of 336 trials. Motion coherence levels were individually adjusted through a titration procedure to obtain an accuracy of 75–80% (average percentage of coherent dots = 58.8%, sd = 22). Active and sham-TMS-only trials and TMS-task trials were randomly intermixed in one experimental session that lasted about 2 h.

*Experiments 3 and 4: dual-site TMS on V5/V1 excitability*. Each participant took part in three experimental sessions on three separate days. The first session served for a careful determination of phosphene sites, i.e., over which a single right

occipital TMS pulse reliably induced left-sided phosphenes, and for a preliminary estimation of Phosphene Threshold (PT). In the following two sessions, after a brief PT reassessment, participants underwent dual-site TMS. During all test blocks, subjects wore earplugs and a blindfold (with their eyes open), while comfortably seated on a chair with their chin on a chinrest to ensure a stable head position.

On each experimental session (second and third day), 5 main experimental and 5 control blocks were run in a randomized order. During each block, dual-coil TMS were delivered at different inter-pulse intervals, randomly chosen from 19 possible delays (30, 45, 60, 75, 90, 105, 120, 135, 150, 165, 180, 195, 210, 225, 240, 255, 270, 285, and 300 ms), while the inter-trial interval was 4 s. In all blocks, the visual test pulse was applied to one of the two occipital regions, right V5 or right V1, according to the group being tested, whereas the conditioning pulse was applied to the right FEF during experimental blocks and over Cz during control blocks. At the end of each block, participants were asked to remove the blindfold to prevent changes in visual cortex excitability by adaptation to darkness or drowsiness. A total of 380 paired pulses were applied, with each session lasting about 3 h.

**TMS**. In all experiments, TMS was applied by means of two high-power Magstim 200[2] machines (Magstim Company, Whitland, UK). Accordingly, the magnetic pulses had a nearly monophasic pulse configuration, with a rise time of ~100 µs, decaying back to zero in about 1 ms. Each stimulator was connected to a double 70 mm standard figure-of-eight coil and was triggered remotely using E-Prime (Psychology Software Tools, Pittsburgh, PA).

In experiments 1 and 2, TMS was applied over right FEF (active and sham) at 65% maximal stimulator output (MSO). This intensity was chosen according to previous studies showing that at this intensity, TMS pulses effectively activate FEF[63] and lead to behavioral changes[64,65]. One TMS machine was used to deliver active TMS pulses, while the second machine was connected to the sham coil. For sham stimulation, the coil was oriented perpendicular to the scalp and positioned just above the coil used for real stimulation.

For experiments 3 and 4, the intensity of the FEF stimulation was the same as in experiment 1 (65% MSO), while the visual test pulse was individually adjusted to sub-threshold intensity for evoking phosphenes (85% phosphene threshold, PT), suited to study modulation in visual cortex excitability by avoiding floor and ceiling effects. PTs were determined for each participant and condition (V5 and V1 stimulation) through a modified binary search algorithm (MOBS), with an upper limit set at 100% MSO and a lower limit at 0%. The initial TMS intensity was set at 50% MSO and then changed by the experimenter according to the participant's response. Since the number of trials required by this algorithm depends on the consistency of the participant's reports, the procedure was considered successful if it converged to a solution within 15 trials. During the threshold assessment, participants were required to report verbally the presence/ absence of phosphenes after each TMS pulse. The average PT over the 2 days was 56 ± 11% and 52 ± 9% of MSO for the V5 group (no difference between experimental sessions: $t_{(5)} = 1.7$, $p = 0.15$) and 52 ± 9 and 52 ± 7% of MSO for the V1 group (no difference between sessions: $t_{(6)} = -0.19$, $p = 0.85$).

**TMS localization**. T1-weighted structural magnetic resonance images (MRI) were acquired with a 3T Siemens Trio Tim scanner (Siemens, Erlangen, Germany). The stimulation site for the right FEF was then individually localized applying the Cortex-based Alignment (CBA) approach in Brain-Voyager QX 2.8 (Brain Innovation, Maastricht, The Netherlands). Briefly, the cortical surface was reconstructed for each participant from the anatomical data and then aligned to an atlas brain that includes the probabilistic group maps of FEF described in ref. [66]. Group maps were back-transformed to the individual brain anatomy and the center of gravity for the area with the highest probability was defined as the TMS target (for a detailed description of CBA, see ref. [66], for details about CBA applied to TMS target localization, see ref. [67]). Finally, to target the FEF and to keep coil position and orientation constant, Talairach coordinates obtained with the CBA were imported to a frameless stereotactic neuro-navigation system (Brainsight; Rogue solution). FEF coordinates were on average (±sd) $x = 27.4 ± 1.7$, $y = -7.9 ± 2.7$, $z = 52.69 ± 4.8$ for experiments 1 and 2, and $x = 27.4 ± 3$, $y = -8.6 ± 3.1$, $z = 50.1 ± 4.7$ for experiments 3 and 4, respectively.

V5 and V1 were defined as scalp sites where TMS could reliably induce moving (V5) or static (V1) phosphenes within the visual field contralateral to the stimulated occipital cortex. The position of each area was stored in Brainsight and used to monitor the coil position within each session and to ensure a consistent coil positioning between sessions. The V5 site (always right hemisphere) was on average (±sd) 4.15 ± 0.52 cm above the inion and 4.33 ± 1.33 cm right of the midline, whereas V1 (also always right hemisphere) was on average 1.66 ± 0.6 cm above the inion and 1.5 ± 0.24 right to the midline. Finally, the control TMS (Cz-Vertex) position was identified according to the 10–20 International system and monitored through the neuro-navigation apparatus.

**TMS-EEG recordings and analysis (experiment 1)**. TMS-compatible EEG equipment (BrainAmp MRplus, BrainProducts) was used for recording EEG activity from the scalp. The EEG was continuously acquired from 61 TMS-compatible Ag/AgCl multitrode electrodes (EasyCap GmbH, Herrsching, Germany) mounted on an elastic cap and positioned according to the 10–10

International System. Additional electrodes were used as ground (TP9) and reference (AFz). The signal was bandpass filtered at 0.1–1000 Hz and digitized at a sampling rate of 5 kHz. Skin/electrode impedance was maintained below 5 KΩ. An additional electrode was positioned on the outer canthus of the left eye to record eye movements (after being referenced to Fp1), whereas horizontal eye movements were detected by referencing AF7 to AF8 offline. To reduce auditory contamination of EEG induced by coil clicks, participants wore earplugs throughout the experiment.

Although EEG was continuously acquired throughout the session, only trials with no visual stimulus (TMS-only) were included in the EEG analysis. The continuous EEG signal was analyzed offline using Brain Vision Analyzer 2.0 (BrainProducts) and Fieldtrip toolbox[68] (http://www.ru.nl/neuroimaging/fieldtrip). All EEG signals were first re-referenced to the average of all electrodes and high-pass filtered at 0.1 Hz (Butterworth zero-phase filter). The large TMS artifact induced by pulse delivery, typically lasting 5–8 ms with our equipment[69], was removed using cubic interpolation for a conservative 15 ms interval following the TMS pulse[70]. Data were then low pass filtered at 85 Hz. A band rejection filter with a bandwidth of 2 Hz was used to remove 50 Hz interference before performing Independent component analysis (ICA) to identify and remove components reflecting residual muscle activity, eye movements, blink-related activity, and residual TMS-related artifacts. The EEG data were then cut into 3-s epochs starting 1500 ms before and ending 1500 ms after the onset of the magnetic stimulus. All segments were visually inspected and removed if still contaminated by residual eye movements, blinks, muscle activity, or TMS-related artifacts that could not be removed by ICA (mean acceptance rate 73%). Remaining trials were down-sampled to 512 Hz.

To assess whether FEF activation could induce a phase realignment over the occipital areas, single-trial data for each condition and each EEG channel were transformed using a Hanning tapered Fast Fourier transform with frequency ranging from 2 to 40 Hz in steps of 1 Hz, with a fixed 500 ms sliding time window moving in steps of 20 ms. The inter-trial phase coherence (ITPC) across trials was calculated for each frequency according to the following formula:

$$ITPC = \left| \frac{1}{N} \sum_{n=1}^{N} \frac{C_n(f)}{|C_n(f)|} \right| \tag{1}$$

Where $C_n(f)$ is the complex Fourier coefficient of trial $n$ of $N$ at frequency $f$ and $||$ indicates the absolute value.

The ITPC values obtained for the attentional and sham condition were compared through cluster-based permutation tests including all channels (61), frequencies (2–40 Hz), and individual time points over a 500 ms time window (from TMS delivery to 500 ms post-pulse) with 2500 permutations and a cluster threshold $p$-value of 0.025. Resulting probabilities were corrected for two-tailed testing. The effect size of the effect of interest (Fig. 1b) was calculated using Cohen's $d$ as implemented in Fieldtrip (ft_freqstatistics with cfg.statistic = 'cohensd') and reported as the average $d$ of the significant occipital cluster, i.e., across O1, Oz, O2, and Iz, 12–18 Hz, 0–200 ms.

**Fluctuation in behavior and visual cortex excitability (experiments 2, 3, and 4)**. All analyses were performed using Matlab (Mathworks Inc., Natick, MA) and the CircStat toolbox. Motion discrimination accuracy was calculated for each interval and then split into two overlapping windows, including 18 intervals each. One window covered the samples between 0 and 200 ms, the other covered the samples from 70.6 to 270.5). Similarly, phosphene perception curves obtained for each condition (FEF/Cz-V5, FEF/Cz-V1) were split into two overlapping windows including 14 intervals each, one covering the samples between 30 and 225 ms and one from 105 to 300 ms.

For each window, participant and condition, accuracy, and phosphene data were linearly detrended to remove linear effects across inter-stimulus intervals and retain any cyclic patterns around the mean. In particular, we wanted to exclude TMS-locked changes in alertness over time that could translate to better performance (higher accuracy or increased phosphene perception rate) for the intervals close to the delivery of the first, FEF TMS pulse and a gradual, slow performance decay for longer intervals. We then tested for the presence of cyclic patterns employing a curve-fitting procedure in custom software using robust nonlinear least-squares fitting. For each condition and participants, we fitted cosine models C(t) between 7 and 25 Hz in 1 Hz steps according to the following formula:

$$C(t) = A \cos(2\pi f t + \varphi) \tag{2}$$

Where $A$, $f$, and $\varphi$ are the amplitude, frequency, and phase of the cosine, respectively. The fitting was performed for each window separately and statistically evaluated R-squared values were extracted for each frequency by means of a bootstrap procedure. Interval labels were randomly permuted over 500 iterations, and all cosine models were fitted to the resulting behavioral pattern each time for each single participant, generating a null distribution of 500 R-squared values. The R-squared values obtained from the actual data were compared to the null distribution to evaluate whether it fell in the top-97.5th percentile. If so, this by definition indicates that the model cosine significantly explained variance in the group data. For phosphene data collected in experiments 3 and 4, we run an additional analysis to test for a significant difference between each main condition

and its control (FEF-V5 vs Cz-V5; FEF-V1 vs Cz-V1). R-squared labels from the two conditions were permuted and the difference of the mean calculated at each iteration. The null and real data were then compared, and real data considered being significant if above the 97.5th percentile. In addition, for each experiment and window of interest, R-squared values were compared by means of a repeated-measures ANOVA with factors Condition (2 levels: FEF-V5 vs Cz-V5 or FEF-V1 vs Cz-V1) and Frequency (19 levels: cosine frequencies from 7 to 25 Hz). To directly test for phase reset for each participant, condition, and frequency, the phase of the best model was extracted and tested for deviation from uniform distribution with the Rayleigh test using CircStat.

The analysis of behavioral and phosphene data was performed on frequencies between 7 and 25 Hz (i.e., in the alpha and beta-band), in order to ensure a broad enough coverage to include the frequency window that showed significant effects in experiment 1 (EEG-effects in a 12–18 Hz frequency window, see "Results"). This also ensured that only oscillatory models that would have at least 1.5 cycles in the tested time window (200 ms) were included in the analysis.

**Tests for the presence of oscillatory activity**. We tested for the presence of oscillatory activity in our signals of interest, as opposed to these signals being explained by non-oscillatory models of brain activity (evoked responses). We applied these tests to both EEG and behavioral data as follows.

*EEG data (experiment 1)*. To test for the presence of oscillatory activity in the brain responses evoked over the occipital channels by right FEF-TMS, we employed the Better OSCillation detection (BOSC) method[26]. The algorithm was applied to the cross-trial averaged signal (TMS-evoked potentials) of those occipital electrodes showing a significant beta phase realignment in the ITPC analysis (results shown in Fig. 1b). Briefly, to identify true oscillatory events in the EEG signal, this method calculates a power threshold from an estimate of the local background activity and a duration threshold that is scaled to each specific frequency. The two thresholds allow to discriminate portions of recorded activity that meet these criteria and reject transient events that are non-oscillatory but that can create changes in power that can be erroneously interpreted as oscillations. The duration threshold was set to three cycles, while the power threshold was estimated for each frequency and electrode as follows. First, wavelet analysis was performed on the entire epoch ($-1.5$ to $1.5$ s, where 0 ms is the time at which TMS was delivered) covering frequencies from 8 to 24 Hz in 17 log-spaced steps. The background spectrum was modeled as colored noise and fits actual power with a linear regression in log–log units. The power threshold for each frequency was calculated as the 95th percentile of the theoretical $\chi^2$ distribution of wavelet power values (for more details about BOSC, see ref. [26]). We then extracted the proportion of time during which signals exceeded the power and duration threshold at a given frequency (indicating the presence of an oscillation) (P-episodes) in the 0–300 ms window after the TMS pulse. This analysis was performed for the real FEF and sham-TMS condition, which were then compared across the frequencies and channels of interest using a cluster corrected two-sided *t*-test.

*Behavioral data (experiments 2, 3, and 4)*. In addition to the oscillatory (cosine) model (see above), we fitted a non-oscillatory model to both behavioral and phosphene time series. This model was constructed to account for the possibility that data were explained by an ERP evoked by the TMS pulse over the occipital areas. The ERP response was modeled as a decaying cosine wave as in ref. [28], according to the following formula:

$$\text{ERP}(t) = A \frac{(t - t_0)}{\tau} e^{-(t - t_0)/\tau} \cos(2\pi f(t - t_0) + \varphi) \tag{3}$$

Where $A$ is the ERP amplitude, $t_0$ is the time at which the ERP deflection starts (the time at which the TMS pulse evokes an activity over the visual area), with $t > t_0$ at any time point, $f$ the frequency and $\varphi$ the phase of the cosine wave. The linear rise and decay constant $\tau$ upper bound was set to 10 ms. As for the oscillatory (cosine) model, this model was fitted for each participant, condition, and window. Furthermore, as $t_0$ was unknown, at each iteration $t_0$ was set to correspond to a tested time point (motion discrimination window1: $t_0$ ranged from 0 to 200 ms in 11.8 ms steps; window2: $t_0$ ranged from 71 to 271 ms in 11.8 ms steps; phosphene data window1: $t_0$ ranged from 30 to 225 ms in 15 ms steps; window2: $t_0$ ranged from 105 to 300 ms in 15-ms steps). Akaike's Information Criterion (AIC) was then used to compare the 2 models, taking into account the respective numbers of fitted (free) parameters per model. The number of free parameters was 4 for the oscillatory (cosine) model (detrend, frequency, phase, amplitude), while there were 6 free parameters for the ERP model, including T0 and $\tau$ in addition to detrend, frequency, phase, and amplitude. For comparisons, per experiment (2, 3, and 4) and participant, the best oscillatory and non-oscillatory models, i.e., yielding the lowest AIC score, were selected among all possible models in the significant frequency range of the cosine fitting results. Following Burnham and Anderson[29], we compared the two models by computing the difference between the model with the highest AIC score and the model with the lowest AIC score and considered an average $\Delta > 9$ as evidence in favor of the better performing model. Note that for all participants and experiments, the cosine models yielded the lowest AIC values. An

example of best fits in a representative participant is shown in Supplementary Fig. 4 (both best-fitting oscillatory and non-oscillatory models shown).

**Reporting summary**. Further information on research design is available in the Nature Research Reporting Summary linked to this article.

## Data availability
The datasets that support the findings of this study are available in Open Science Framework (OSF) at the following permanent link: osf.io/nx3yv. Source data are provided with this paper.

## Code availability
Custom codes used to analyze the data are available in OSF at the following permanent link: osf.io/nx3yv.

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

## Acknowledgements

This work was supported by a Wellcome Trust Award to Gregor Thut and Joachim Gross [Grant number 098434, 098433]. Joachim Gross was further supported by the DFG (GR 2024/5-1) and IZKF Münster (Gro3/001/19). We would like to thank Catriona Thompson, Hillary Brown, and Alessandra Vergallito for helping with data collection, Luigi D'Auria for advice on the ERP model, and Chris Madan for useful discussions about the BOSC method.

## Author contributions

D.V. and G.T. conceived and designed the study. D.V., J.G., and G.T. performed the E.E.G. analysis. F.D., A.T.S., and D.V. implemented the T.M.S. localization procedure. D.V., J.G., and S.M. analyzed behavioral data. D.V. performed the experiments. D.V. and G.T. wrote the paper. All authors discussed results and implications and commented on the manuscript at all stages.

## Competing interests

The authors declare no competing interests.
