## [Peer Review File · Nature Communications]

Reviewers' comments:

Reviewer #1 (Remarks to the Author):

This is a potentially interesting study using TMS-EEG and dual-site TMS to test the effects of emulation of an attentional signal in the frontal eye field (FEF) by single-pulse TMS on oscillatory phase re-alignment in down-stream visual areas and its behavioral consequences on static and moving phosphene thresholds. They find phase re-alignment as indexed by an increase in inter-trial phase coherence (ITPC) over occipital electrodes in the high-alpha low-beta band. Dual-site FEF-V5 stimulation resulted in modulation of the rate of perceived moving phosphenes, best modelled by a cosine function at 16-17 Hz frequency. No such modulation of phosphene perception was observed in the FEF-V1 experiment. They concluded that top-down signals originating in FEF causally shape visual cortex activity through mechanisms of oscillatory realignment.

Major comments:

1. Experiment 1: While the auditory click has been controlled by a sham stimulation condition, TMS results also in evoked EEG responses through activation of somatosensory input from scalp stimulation [e.g., Rogasch et al. 2014, *NeuroImage* 101:425-439], and this has not been controlled for. Therefore, it cannot be entirely excluded that the increase in ITPC was caused non-specifically by input from somatosensory afferents rather than TMS-induced activation of the brain. To demonstrate specificity of FEF activation for causing the phase reset over occipital electrodes, it would be mandatory to either use a sham condition that also controls for activation of somatosensory afferents [e.g., Gordon et al. 2018, *Brain Stimul* 10.1016/j.brs.2018.08.003], or a real TMS condition over a control site.
2. The very low number of subjects in both experiments (n=6 in Experiment 1, n=7 each for the V5 and V1 conditions in Experiment 2) is a major concern. How can the authors be sure that the reported findings are not spurious? Furthermore, in Experiment 2, only females were tested. This potentially limits generalizability of the reported findings.
3. While the phase reset (increase in ITPC) in Experiment 1 is reported over occipital electrodes (Figure 1), i.e., probably including primary and secondary visual cortices, Experiment 2 revealed a dissociation (significant cyclic modulation of excitability only in V5 but not V1). Albeit the authors discuss possible reasons, it would be of interest to shed more light on this apparent discrepancy based on the obtained EEG data in Experiment 1. Would it be possible to analyze the ITPC data on the individual source level to understand better which parts of the occipital cortex show phase reset (i.e., V5 vs. V1)?
4. Figure 3A suggests that there is also a strong trend for an R-squared difference between V5 and Cz control conditions in the frequency range of 10-12 Hz (negative values). These may just not be significant due to low sample size and 1 outlier (with a positive difference). If this difference became significant (when testing more subjects, and/or eliminating outliers) then this would suggest a significant cyclic modulation effect of the control condition (Cz stimulation) on V5 excitability.
5. The theory of communication through coherence (Fries 2005, *Trends Cogn Sci* 9:474-480) predicts that the effects of FEF stimulation on excitability in other nodes of the network depend on phase synchronicity and excitability (i.e., phase of the local oscillations). To demonstrate this in the current experiments would strengthen the data but will require real-time EEG estimation of phase and phase-synchronicity of the stimulated sites (FEF, V5, V1). This way, it may also become possible to demonstrate top-down effects also from FEF to V1.
6. Novelty of the methodological approach is limited. Very similar dual-site TMS experiments have been obtained by an overlapping group of authors previously in the motor cortex (Picazio et al. 2014, *Curr Biol* 24:2940-2945).

Other specific comments:

1. Typos in legends of Figs. 2 and 4: excitabilty > excitability.
2. Typo p. 25: 5 K Ω > 5 k Ω
3. Typo p. 27: 1H Hz step > 1Hz step

4. Fig. 3B: Units in the phase plots are unclear. Are these unity plots with a maximum value of 1?
5. Figs. 2A-B and Figs. 4A-B: It is not fully clear what is shown on the y-axes. Is this phosphene perception rate normalized to the mean of all interstimulus intervals? What were the mean absolute perception rates?
6. The authors stated: "This study was part of a larger experiment, during which participants were asked to perform a simple motion detection task. Only trials with no visual stimulus were included in the present analysis." This is somewhat irritating, as no further specific information is provided as to the details of the "larger experiment", and how motion detection trials were intermingled with trials without visual stimulus.
7. Methods: The authors stated: "To reduce auditory contamination of EEG induced by coil clicks, participants wore earplugs throughout the experiment." This is not state-of-the-art. To reduce auditory-evoked activity in the EEG, masking white noise should be used (e.g., Massimini et al. 2005, *Science* 309:2228-2232).

Reviewer #2 (Remarks to the Author):

In this study, Veniero et al aim to investigate the causal role of causal FEF on perceptual sampling. In Experiment 1, the authors used transcranial magnetic stimulation (TMS) in frontal eye fields (FEF) as a 'reset event', and analyse the effect of this reset on the whole brain oscillatory pattern. They found that FEF stimulation was followed by an increase in phase consistency around 15Hz. This oscillatory modulation was more pronounced over occipital sites, and lasted for approximately 200ms. Then, in a second experiment, the authors applied a second TMS over visual areas at different intervals following the 'reset' FEF pulse. Using this protocol, the authors investigated the temporal pattern of phosphene induced by the stimulation of visual areas, immediately after FEF stimulation. Their results show that phosphenes induced by V5, but not V1, stimulation were modulated by the FEF TMS pulse. Similar to the findings in Experiment 1, they observed that phosphene perception fluctuated in the beta band (~16Hz), locked to the FEF stimulation. Together, these results show that top-down signals, emerging from FEF, modulate visual perception in a rhythmic manner. This is an interesting finding, with relevant implications for current theories of attention and perception.

My main concern with this study is with small sample size (6 subjects in Exp1 and 7 subjects in each study in Exp2). This, combined with the relative small number of trials in Experiment 2 (20 trials per interval), results in a very weak statistical sensitivity. Unfortunately, the only way to solve this problem would be to increase the sample size.

The authors should also cite and discuss the potential discrepancies with the findings reported in Marshall et al (2015). I understand the differences between using single pulse TMS and theta-burst rTMS. However, I do not believe that they could account for the differences in lateralization of the effects.

Marshall, T. R., O'Shea, J., Jensen, O., & Bergmann, T. O. (2015). Frontal eye fields control attentional modulation of alpha and gamma oscillations in contralateral occipitoparietal cortex. *Journal of Neuroscience*, 35(4), 1638–1647.

Reviewer #3 (Remarks to the Author):

Top-down signals from the Frontal Eye Fields modulate visual cortex excitability by phase-

realignment of oscillatory beta activity

Domenica Veniero, Joachim Gross, Stephanie Morand, Felix Duecker, Alexander Sack, and Gregor Thut

Previous work has shown the phase of ongoing neural oscillations impacts perception. However, this work has largely relied on correlations between oscillation phase and perceptual sensitivity/accuracy. In the current manuscript, Veniero et al causally manipulate top-down control signals, showing single pulse transcranial magnetic stimulation (spTMS) to human frontal eye fields (FEF) induces phase-resetting in visual cortex. Then, using a second spTMS over visual cortex, the authors test whether the induced top-down control signals increases the likelihood of seeing an induced phosphene and whether this likelihood varies in a cyclic manner. Overall, this is a short but interesting paper. Although not completely novel (see comments below), the results are timely, and the causal manipulation will add to the literature on beta oscillations. However, I have a few concerns:

1) The authors test for oscillatory modulation of perception by fitting a cosine to the likelihood of detecting a phosphene. However, this assumes oscillations exist in the behavioral data. Did the authors compare the cosine model to any other non-oscillatory model? For example, a constant or exponential decay model (simpler) or a slightly more complicated model, such as modeling an evoked potential (e.g. as a decaying sinusoid) might fit better. These would have different interpretations.

For example, one could explain the effects as an evoked potential with some power in the beta frequency band. This would fit with the fact that the authors show the oscillatory effect on phosphene perception drops off quickly (not existing after the first 100 ms). While still arguing for a role of FEF in top-down control it would lead to a different interpretation than a sustained oscillation.

The authors should provide model comparison statistics for tests against alternative models.

2) Related to this, the authors argue that there is significant phase-locking of responses across subjects in the FEF-TMS condition (Figure 3B). However, it seems that this comparison is done to baseline (i.e. zero phase consistency), when it should be compared against the control stimulation condition (Cz-TMS) to appropriately control for any general effects of TMS stimulation.

3) Also related to comment #1, what did the raw evoked potential look like over occipital cortex (i.e. before the time-frequency decomposition in Figure 1). Did it look like a sustained oscillation? This is a particular concern given the long time windows used to estimate phase (500 ms), which could spread out an impulse response over a broader time window.

4) The authors argue that causal drive of FEF leads to phase-resetting and that this modulates perception. However, the effect of attention on detection tasks is known to change both the perceptual sensitivity of subjects (d') as well as the threshold of detection (criterion). The current approach does not allow the authors to distinguish between these hypotheses (or if both occur). This could be addressed by adding a more complex behavioral task that would be sensitive to either change.

5) The authors do a thorough job of citing the relevant research around oscillations and attention, with a particular focus on the systems neuroscience literature. While I think this is appropriate, I felt that some of the existing literature on the effects of TMS (either single-pulse or repetitive) on oscillations in visual cortex was a bit light. Perhaps this stood out to me as the authors play a leading role in this literature. In particular, I thought the existing literature on alpha (~ 10 Hz) rhythms was missing. Furthermore, previous single pulse studies in FEF have shown an increase in perception time-locked to FEF stimulation (e.g. Grosbras and Paus, EJM 2003; also at ~ 10 Hz).

6) Related to this, a lot of previous work has shown TMS of FEF modulates alpha-band rhythms in occipital cortex. So, it is a bit surprising that the current experiment does not show any such modulation. Do the authors have an explanation for this discrepancy? I feel this should at least be discussed.

7) Previous psychophysical work has shown perceptual modulation from stimulus onset is at much lower frequencies (e.g. \sim theta in Landau et al). From Figure 3 and 4 it seems as if the authors focused on higher frequency rhythms (> 7 Hz). Did the authors test lower frequencies? In particular, there appears to be some late phase alignment at \sim 4-5Hz in Figure 1B and perhaps some modulation of phosphene detection from V1-TMS at \sim 5 Hz in Figure 4B (oddly maybe greater in the control condition?).

8) Related to comment #7, the authors note previous work showing phase alignment of behavioral responses to sudden onset visual stimuli in the discussion. However, they gloss over the fact that this occurs at a much different frequency. I think there are very reasonable explanations for this discrepancy (e.g. nesting of oscillations) but this should at least be noted and discussed.

9) The authors conclude that the period of phase alignment in EEG (Figure 1) coincides with the effects on phosphene perception (Figures 2 and 3). However, this isn't explicitly shown – what is the phase of the beta oscillation seen in Figure 1? Does it generally coincide with the expected phase given the periodicity in excitability seen with TMS of visual cortex?

10) Do the authors control for the number of comparisons across frequency (e.g. Figure 3A)?

11) R-squared is bounded and therefore highly nonlinear, making it difficult interpret differences, such as the quality of fit of cosine model for FEF-TMS and Cz-TMS (e.g. Figure 3A). The authors do an appropriate permutation test that should partially control for this issue, but did the authors also test other statistics of model fit (e.g. log-likelihood or percent explained variance)?

12) Related to this, what were the raw R-squared for the different frequency cosine models in the attention and control conditions? Was there a peak at the 16-17 Hz in the attention condition?

Reviewer #1 (Remarks to the Author):

This is a potentially interesting study using TMS-EEG and dual-site TMS to test the effects of emulation of an attentional signal in the frontal eye field (FEF) by single-pulse TMS on oscillatory phase re-alignment in down-stream visual areas and its behavioral consequences on static and moving phosphene thresholds. They find phase re-alignment as indexed by an increase in inter-trial phase coherence (ITPC) over occipital electrodes in the high-alpha low-beta band. Dual-site FEF-V5 stimulation resulted in modulation of the rate of perceived moving phosphenes, best modelled by a cosine function at 16-17 Hz frequency. No such modulation of phosphene perception was observed in the FEF-V1 experiment. They concluded that top-down signals originating in FEF causally shape visual cortex activity through mechanisms of oscillatory realignment.

R: we would like to thank the reviewer for his/her comments that helped us improve the original manuscript. We would like to start by pointing out that the original dataset has been extended to include more subjects and an additional experiment, which is now a central part of the manuscript. Specifically, and relevant for some of this reviewer's comments, the number of participants recruited for the EEG part has been doubled and a new experimental condition has been added, in which participants were asked to perform a motion discrimination task following the activation of the right FEF. The results of this additional experiment are in line with the phosphene data and show that the activation of the FEF causes the behavioural performance - quantified as accuracy in motion discrimination - to oscillate over time with a beta rhythm. In light of these changes and new results, we hope that this reviewer will find our manuscript improved and its interest/contribution substantiated.

Major comments:

1. Experiment 1: While the auditory click has been controlled by a sham stimulation condition, TMS results also in evoked EEG responses through activation of somatosensory input from scalp stimulation [e.g., Rogasch et al. 2014, *NeuroImage* 101:425-439], and this has not been controlled for. Therefore, it cannot be entirely excluded that the increase in ITPC was caused non-specifically by input from somatosensory afferents rather than TMS-induced activation of the brain. To demonstrate specificity of FEF activation for causing the phase reset over occipital electrodes, it would be mandatory to either use a sham condition that also controls for activation of somatosensory afferents [e.g., Gordon et al. 2018, *Brain Stimul* 10.1016/j.brs.2018.08.003], or a real TMS condition over a control site.

R: We agree with the reviewer that, as recently demonstrated in several papers, TMS generates somatosensory responses that represent an artifactual source in the EEG signal. We also acknowledge that in our TMS-EEG experiment (one of 4 experiments), we have not controlled for this possible confound. However, in light of the collective results of all four experiments, some of which include a real TMS condition over a control site (Cz), we can confidently exclude that the ITPC enhancement simply reflects non-specific somatosensory input.

Accordingly, the results of the EEG-TMS experiment represent a starting point indicating a beta activity being elicited by FEF-control over the occipital cortex. We sought confirmation of this result with a series of follow-up experiments that one by one investigated different alternative explanations of our findings. Namely, we investigated the behavioural consequences of the induced beta oscillation to firmly link the occipital EEG oscillation to visual function, and also

checked that effects are not observed after TMS of a non-FEF site (using real TMS over Cz as control) to firmly link the visual effect to FEF-stimulation rather than somatosensory confounds.

More specifically, the EEG results (experiment 1) point to a FEF-triggered phase alignment in the beta band over the occipital electrodes. We reasoned that if this is a real effect of FEF-activation on visual cortex activity, the ability to perform a visual task should also change accordingly. Therefore, we asked our participants to discriminate the direction of random dots that were presented at different time delays after FEF-activation (experiment 2). As shown by the results of this experiment, the discrimination accuracy also fluctuated over time and this fluctuation was explained by a beta oscillation. This hence more firmly links the FEF-TMS triggered occipital EEG oscillation to a visual function, and hence visual cortex activity and rules out the possibility that this beta activity reflects spurious activation of somatosensory areas. To further corroborate our findings, we then planned two more follow-up experiments (experiments 3 and 4), this time in an independent sample. We directly stimulated visual areas (V1 or V5) with a second TMS coil to test their excitability changes (by phosphene induction) after different delays from the FEF-activation (further testing visual cortex implication in the FEF-TMS triggered beta oscillation), and including a real TMS site (Cz-TMS) as control. We found, again, that the excitability of the visual cortex oscillates with a beta frequency, but only when the effects of FEF-stimulation are measured over V5 but not V1. Importantly, we also checked that the fluctuation of phosphene perception rate in the V5/V1 condition could not be explained by somatosensory confounds by investigating phosphene perception after Cz (instead of FEF) stimulation. In conclusion, our collective results provide strong evidence that the beta activity is generated within the extrastriate visual cortex (V5) and that this activity is directly related to FEF-activation and visual perception and not to somatosensory input.

This is now more clearly outlined in the discussion section.

2. The very low number of subjects in both experiments (n=6 in Experiment 1, n=7 each for the V5 and V1 conditions in Experiment 2) is a major concern. How can the authors be sure that the reported findings are not spurious? Furthermore, in Experiment 2, only females were tested. This potentially limits generalizability of the reported findings.

R: As mentioned above, we have doubled the number of participants and run a whole new experiment. Over all four experiments, we now confirm the presence of FEF-TMS triggered beta oscillations in visual areas/ visual function in a total of 25 participants. Unfortunately, the number of men remains low (3 participants) as our subject pool mainly consists of psychology students, a population characterised by a significant gender imbalance.

In our case, putative gender differences might have affected our results in the following 3 ways. First, there may be gender differences in cortical excitability and therefore how men and women react to TMS. However, a recent study published in *Clinical Neurophysiology* (2016) has addressed this issue in a sample of 200 participants (100 males, 100 females) and found no difference in a wide range of excitability indexes, such as motor threshold, suprathreshold MEPs, ICF and SICI (Cueva et al, 2016). More closely related to our study, Esterman et al. (2015) assessed the role of left and right FEF in sustained attention through rTMS @1Hz and found that the behavioural effects did not change as a function of gender. Second, there may be gender differences in brain connectivity/anatomy affecting our results. The most common finding is that women show more inter-hemispheric connectivity, while men have stronger intra-hemispheric connectivity

(Ingalhalikar et al., 2014). However, some studies comparing connectivity measures found that differences between men and women become small or non-significant when comparing participants with similar brain sizes (e.g. Hänggi et al., 2014). Furthermore, Joel and collaborators (2015) provided evidence that seriously challenges the idea of a sexual dimorphism when looking at the human brain. After analysing 1,400 human brains from four datasets, they showed that the female and male distributions for grey matter, white matter, and connections are greatly overlapping such that they rejected the idea of a male vs female brain. Also, they pointed out other factors than gender that might better predict individual neural differences, such as learning experiences. Finally, men and women might be different in terms of brain dynamics which could affect our results. A few studies are available and some of them report opposite findings. However, a study examining MEG data from 77 participants during a visuospatial task (Wiesman et al., 2019) has shown no gender difference in any frequency band except for gamma generated within the superior parietal cortex. Hence, while the question of whether gender differences are significantly shaping the human brain is an interesting, yet controversial topic, we do not think that the gender bias in our sample poses problems for the interpretability and generalizability of our results (as other variables, such as culture, age, education, gene pool and skills may be more relevant than gender).

References

Cueva AS, Galhardoni R, Cury RG, Parravano DC, Correa G, Araujo H, Cecilio SB, Raicher I, Toledo D, Silva V, Marcolin MA, Teixeira MJ, Ciampi de Andrade D. Normative data of cortical excitability measurements obtained by transcranial magnetic stimulation in healthy subjects. *Neurophysiol Clin*. 2016. <http://10.1016/j.neucli.2015.12.003>. Epub 2016 Feb 26

Esterman M, Liu G, Okabe H, Reagan A, Thai M, DeGutis J. Frontal eye field involvement in sustaining visual attention: evidence from transcranial magnetic stimulation. *Neuroimage*. 2015; <https://doi.org/10.1016/j.neuroimage.2015.01.044>.

Ingalhalikar M, Smith A, Parker D, Satterthwaite TD, Elliott MA, Ruparel J, Hakonarson H, Gur RE, Gur RC, Verma R. Sex differences in the structural connectome of the human brain. *PNAS*, 2014; <https://doi.org/10.1073/pnas.1316909110>.

Hänggi J, Fövényi L, Liem F, Meyer M, Jäncke L. The hypothesis of neuronal interconnectivity as a function of brain size—a general organization principle of the human connectome. *Front Hum Neurosci*. 2014 Nov 11;8:915. <https://doi.org/10.3389/fnhum.2014.00915>.

Joel D, Berman Z, Tavor I, Wexler N, Gaber O, Stein Y, Shefi N, Pool J, Urchs S, Margulies DS, Liem F, Hänggi J, Jäncke L, Assaf J. Sex beyond the genitalia: The human brain mosaic. *PNAS* 2015; <https://doi.org/10.1073/pnas.1509654112>.

Wiesman AI and Wilson TW. The Impact of Age and Sex on the Oscillatory Dynamics of Visuospatial Processing. *PNAS* 2019 <https://doi.org/10.1016/j.neuroimage.2018.10.036>

3. While the phase reset (increase in ITPC) in Experiment 1 is reported over occipital electrodes (Figure 1), i.e., probably including primary and secondary visual cortices, Experiment 2 revealed a dissociation (significant cyclic modulation of excitability only in V5 but not V1). Albeit the authors discuss possible reasons, it would be of interest to shed more light on this apparent discrepancy based on the obtained EEG data in Experiment 1. Would it be possible to analyze the ITPC data on the individual source level to understand better which parts of the occipital cortex show phase reset (i.e., V5 vs. V1)?`

R: Our TMS-EEG experiment was not designed to perform a source analysis. We only have 61 channels and their positions were not co-localized to anatomical space within each participant, nor do we have information on individual anatomical V1/V5-coordinates. The only way to perform a source analysis would require choosing an alternative way of defining these coordinates, with the results being highly dependent on this choice. Furthermore, V1 and V5 are close and EEG in general might fail in separating the activity these two areas generate, even more so considering the limited information we have, i.e. no individual V1/V5 coordinates and electrode positions in anatomical space. Therefore, we do not feel that source analysis would yield reliable results.

Nonetheless, we have attempted to analyse the ITPC at the source level using average coordinates reported in Plomp et al. (2015), where the right and left visual areas were functionally localized for each participant using fMRI. The EEG signal was source projected and the activity corresponding to V1 and V5 (Left V5=-48 -69 7; Right V5= 50 -66 11; Left V1 -14 -100 7; Right V1= 17 -97 9) was extracted for each participant and condition, i.e. real and sham. The ITPC was then calculated as for the scalp EEG signal in experiment 1, i.e. extracted for each condition and then sham-ITPC subtracted from the real-ITPC. Finally, we run a cluster corrected permutation statistic to test for difference across the 4 regions. As shown in the figure below, the source-projected ITPC difference between real and sham TMS was strongest in right V5 (RV5) relative to left and right V1 (LV1, RV1) and left V5 (LV5), but when correcting for multiple comparisons there is no difference between V1 and V5 activity within each hemisphere.

Due to the abovementioned technical limitations for source reconstruction of EEG in general and of our design in particular (no individual coordinates, no electrode positions, low number of channels), we would prefer not to include this analysis in the manuscript. However, please note that in the follow-up experiments 3&4, we leverage on the spatial specificity of TMS to make a point about where in the occipital regions the beta activity is generated. Using occipital TMS to induce static versus moving phosphenes, we can functionally localize the two visual areas reliably, and therefore identify which of these areas shows a phase reset in excitability, as tested by the difference between phosphene perception curves in response to TMS over V1 and V5.

References:

Plomp G, Hervais-Adelman A, Astolfi L, Michel CM. Early recurrence and ongoing parietal driving during elementary visual processing. *Sci Rep* (2016): <https://doi.org/10.1038/srep18733>

4. Figure 3A suggests that there is also a strong trend for an R-squared difference between V5 and Cz control conditions in the frequency range of 10-12 Hz (negative values). These may just not be significant due to low sample size and 1 outlier (with a positive difference). If this difference became significant (when testing more subjects, and/or eliminating outliers) then this would suggest a significant cyclic modulation effect of the control condition (Cz stimulation) on V5 excitability.

R: We agree with the reviewer that collecting more data was crucial. We have decided to further test for fluctuations in behaviour by collecting data during a behavioural task, namely a motion discrimination task rather than adding participants to the phosphene data. In summary, our collective data over all 4 experiments strongly converge to reveal significant effects in the beta band, while none points to significant effects in the 10-12Hz range, or corroborated the trend in this range as seen in the V5 data only.

It is conceivable that the Cz condition picks up another cyclic modulation, namely in the alpha band over visual areas, which could be induced by the TMS-associated sound and as demonstrated for example by us in Romei et al. (2012). However, this alpha-band modulation does not significantly discriminate between sham and real TMS which share the same auditory stimulation, and is also not picked up in any of the other experiments (including the new behavioural data of experiment 2).

Altogether, we are therefore confident that the observed beta activity reflects the main effect of FEF-induced (top-down controlled) phase modulations in the visual cortex.

References

Plomp G, Hervais-Adelman A, Astolfi L, Michel CM. Early recurrence and ongoing parietal driving during elementary visual processing. *Sci Rep* (2016). <https://doi.org/10.1038/srep18733>

Romei V, Gross J, Thut G. Sounds Reset Rhythms of Visual Cortex and Corresponding Human Visual Perception. *Curr Biol* 2012; <https://doi.org/10.1016/j.cub.2012.03.025>

5. The theory of communication through coherence (Fries 2005, *Trends Cogn Sci* 9:474-480) predicts that the effects of FEF stimulation on excitability in other nodes of the network depend on phase synchronicity and excitability (i.e., phase of the local oscillations). To demonstrate this in the current experiments would strengthen the data but will require real-time EEG estimation of phase and phase-synchronicity of the stimulated sites (FEF, V5, V1). This way, it may also become possible to demonstrate top-down effects also from FEF to V1.

R: We thank the reviewer for this suggestion. We agree that it would be interesting to see if FEF control over lower-level visual areas depends on phase synchrony. Addressing this however would represent a different study (testing the CTC theory was not our aim) and would also require a different design/methodology allowing for more accurate source localization.

6. Novelty of the methodological approach is limited. Very similar dual-site TMS experiments have been obtained by an overlapping group of authors previously in the motor cortex (Picazio et al. 2014, *Curr Biol* 24:2940-2945).

R: We agree that the methods we use are not novel. For this reason, we never stress(ed) methodological innovations in our paper. Yet, we believe that our results have a high novelty value, illustrating for the first time how an attentional impulse over FEF (emulated by FEF-TMS) reorganizes perceptually relevant oscillatory activity over visual areas, and this now more robustly after adding more participants and an additional behavioural experiment.

Other specific comments:

1. Typos in legends of Figs. 2 and 4: excitabilty > excitability.

2. Typo p. 25: 5 K Ω > 5 k Ω

3. Typo p. 27: 1H Hz step > 1Hz step

R: 1-3, corrected.

4. Fig. 3B: Units in the phase plots are unclear. Are these unity plots with a maximum value of 1?

R: Figure 3B has been removed. All polar plots show the data as points on the unit circle.

5. Figs. 2A-B and Figs. 4A-B: It is not fully clear what is shown on the y-axes. Is this phosphene perception rate normalized to the mean of all interstimulus intervals? What were the mean absolute perception rates?

R: Percentage of perceived phosphenes (experiments 3 and 4) and discrimination accuracy (experiment 2) were linearly detrended to remove linear effects across inter-stimulus intervals and retain any cyclic patterns around the mean. This was stated in the caption but, following the reviewer's comment, we now also changed the y-axis label in Figures 3, 4, and 6 to further clarify that data were detrended.

6. The authors stated: "This study was part of a larger experiment, during which participants were asked to perform a simple motion detection task. Only trials with no visual stimulus were included in the present analysis." This is somewhat irritating, as no further specific information is provided as to the details of the "larger experiment", and how motion detection trials were intermingled with trials without visual stimulus.

R: The larger experiment has now been completed and included in the manuscript (experiment 2). A full description is provided.

7. Methods: The authors stated: "To reduce auditory contamination of EEG induced by coil clicks, participants wore earplugs throughout the experiment." This is not state-of-the-art. To reduced auditory-evoked activity in the EEG, masking white noise should be used (e.g., Massimini et al. 2005, Science 309:2228-2232).

R: We understand the reviewer's comment. However, the use of a sham TMS to control for the auditory artifact is a valid alternative to the white noise approach used by others. To the best of our knowledge, there is no evidence that one approach is superior to the other. Our choice for sham TMS is based on the discomfort that participants report when the white noise is played for a long time, but most importantly on the impossibility to mask completely the coil click because of bone conduction. Applying sham stimulation with the same intensity as the real stimulation, together with a real TMS control (e.g. Cz), gives the experimenter more control over the auditory artifact which can be eliminated by subtraction, whereas with white noise there is no control over how much of the auditory signals contaminate the EEG data, because of the bone conduction.

Reviewer #2 (Remarks to the Author):

In this study, Veniero et al aim to investigate the causal role of causal FEF on perceptual sampling. In Experiment 1, the authors used transcranial magnetic stimulation (TMS) in frontal eye fields (FEF) as a 'reset event', and analyse the effect of this reset on the whole brain oscillatory pattern. They found that FEF stimulation was followed by an increase in phase consistency around 15Hz. This oscillatory modulation was more pronounced over occipital sites, and lasted for approximately 200ms. Then, in a second experiment, the authors applied a second TMS over visual areas at different intervals following the 'reset' FEF pulse. Using this protocol, the authors investigated the temporal pattern of phosphene induced by the stimulation of visual areas, immediately after FEF stimulation. Their results show that phosphenes induced by V5, but not V1, stimulation were modulated by the FEF TMS pulse. Similar to the findings in Experiment 1, they observed that phosphene perception fluctuated in the beta band (~16Hz), locked to the FEF stimulation. Together, these results show that top-down signals, emerging from FEF, modulate visual perception in a rhythmic manner. This is an interesting finding, with relevant implications for current theories of attention and perception.

R: we would like to thank the reviewer for the positive comments about our findings.

My main concern with this study is with small sample size (6 subjects in Exp1 and 7 subjects in each study in Exp2). This, combined with the relative small number of trials in Experiment 2 (20 trials per interval), results in a very weak statistical sensitivity. Unfortunately, the only way to solve this problem would be to increase the sample size.

R: We thank the reviewer for these comments, in response to which we have now collected more data. The number of participants recruited for the EEG experiment has been doubled. We are also including an additional behavioural experiment that complements our previous data. In the original version of the manuscript, we reported changes in visual cortex excitability after different delays from FEF stimulation as measured by the percentage of perceived phosphenes to occipital TMS. In the current version, we show in addition that fluctuations in visual perception is also evident when participants are presented with real visual stimuli, rather than illusory percepts. Specifically, participants were asked to perform a visual motion discrimination task, with motion stimuli presented at different time delays after FEF activation. The accuracy of their performance again fluctuated over time and was explained by a beta oscillation.

The authors should also cite and discuss the potential discrepancies with the findings reported in Marshall et al (2015). I understand the differences between using single pulse TMS and theta-burst rTMS. However, I do not believe that they could account for the differences in lateralization of the effects.

Marshall, T. R., O'Shea, J., Jensen, O., & Bergmann, T. O. (2015). Frontal eye fields control attentional modulation of alpha and gamma oscillations in contralateral occipitoparietal cortex. *Journal of Neuroscience*, 35(4), 1638–1647.

R: The study by Marshall et al. shows modulation of occipital alpha contralateral to the stimulated FEF. When they stimulated the right FEF using TMS with continuous theta burst stimulation (cTBS), attention-related alpha-modulation was reduced in the left visual cortex and vice versa, left FEF cTBS modulated right occipital alpha. While the difference in TMS protocol and task used might explain the difference in frequency found in the present study relative to Marshall et al., we agree with the reviewer that the differences in lateralization may be more difficult to reconcile. However, as reported by Marshall et al., there is a large amount of evidence that supports the idea that the right FEF can modulate activity of both the right and left posterior areas. This has been

shown with single-pulse TMS and behavioural measures (Grosbras et al., 2002), paired-pulse TMS and phosphenes (Silvanto et al., 2006), rTMS and BOLD signal (Heinen., et al 2013), rTMS and EEG (Capotosto et al, 2009), as well as cTBS and behavioural measures (Duecker et al., 2013). We have included a brief discussion of this and related papers in the discussion section.

References:

Grosbras MH and Paus T. Transcranial Magnetic Stimulation of the Human Frontal Eye Field: Effects on Visual Perception and Attention. J Cogn Neurosci 2002; <https://doi.org/10.1162/089892902320474553>

Silvanto J, Lavie N, Walsh. Stimulation of the human frontal eye fields modulates sensitivity of extrastriate visual cortex. J Neurophysiol 2016; <https://doi.org/10.1152/jn.00015.2006>

Capotosto P, Babiloni C, Romani GL, Corbetta M. Frontoparietal cortex controls spatial attention through modulation of anticipatory alpha rhythms. J Neurosci 2009; <https://doi.org/10.1523/JNEUROSCI.0539-09.2009>

Heinen K, Feredoes E, Weiskopf N, Ruff CC, Driver J. Direct evidence for attention-dependent influences of the frontal eye-fields on feature-responsive visual cortex. Cereb Cortex 2014; <http://doi.org/10.1093/cercor/bht157>

Duecker F, Formisano E, Sack AT. Hemispheric differences in the voluntary control of spatial attention: direct evidence for a righthemispheric dominance within frontal cortex. J Cogn Neurosci 2013; https://doi.org/10.1162/jocn_a_00402

Reviewer #3 (Remarks to the Author):

Top-down signals from the Frontal Eye Fields modulate visual cortex excitability by phase-realignment of oscillatory beta activity

Domenica Veniero, Joachim Gross, Stephanie Morand, Felix Duecker, Alexander Sack, and Gregor Thut

Previous work has shown the phase of ongoing neural oscillations impacts perception. However, this work has largely relied on correlations between oscillation phase and perceptual sensitivity/accuracy. In the current manuscript, Veniero et al causally manipulate top-down control signals, showing single pulse transcranial magnetic stimulation (spTMS) to human frontal eye fields (FEF) induces phase-resetting in visual cortex. Then, using a second spTMS over visual cortex, the authors test whether the induced top-down control signals increases the likelihood of seeing an induced phosphene and whether this likelihood varies in a cyclic manner. Overall, this is a short but interesting paper. Although not completely novel (see comments below), the results are timely, and the causal manipulation will add to the literature on beta oscillations. However, I have a few concerns:

R: We thank the reviewer for her/his comments, which helped us improve the quality of the paper and strengthen our results. As the reviewer will see, we have collected data from additional participants for the EEG experiment and have also included a new experiment whereby participants were asked to perform a motion discrimination task (experiment 2). Moving dots were presented at different intervals after TMS over FEF. Similarly to the EEG results and

phosphene data testing visual cortex excitability after FEF-TMS, we found that performance accuracy in this task fluctuates over time at a beta frequency, thus confirming our original results and conclusion that FEF activation phase-resets the activity of the visual cortex (now experiments 1, 3 and 4). We hope this reviewer finds the new version of the manuscript improved.

1) The authors test for oscillatory modulation of perception by fitting a cosine to the likelihood of detecting a phosphene. However, this assumes oscillations exist in the behavioural data. Did the authors compare the cosine model to any other non-oscillatory model? For example, a constant or exponential decay model (simpler) or a slightly more complicated model, such as modelling an evoked potential (e.g. as a decaying sinusoid) might fit better. These would have different interpretations.

For example, one could explain the effects as an evoked potential with some power in the beta frequency band. This would fit with the fact that the authors show the oscillatory effect on phosphene perception drops off quickly (not existing after the first 100 ms). While still arguing for a role of FEF in top-down control it would lead to a different interpretation than a sustained oscillation.

The authors should provide model comparison statistics for tests against alternative models.

R: We thank the reviewer for this very valuable suggestion. We have run additional analyses to fit other non-oscillatory models to the phosphene data and the new behavioural data, for comparison to our original oscillatory (cosine) model. As additional models, we employed an exponential decay and a decaying sinusoid, as previously used to model an ERP (as in Mazaheri and Jensen, 2006). To evaluate the significance of the fit to the data, we applied a bootstrap procedure per alternative model (as for the examination of the cosine model, see text). In addition, we directly compared all three models in terms of goodness of fit to the data. Our new analyses revealed that only the cosine models in the beta band significantly explained behavioural and phosphene fluctuations over time. Importantly, a repeated measure Anova on the R-squared values obtained for each model revealed that the oscillatory (cosine) model explains more variance as compared to the non-oscillatory (exponential and decaying sinusoid) models for both experiments (motion discrimination and phosphene data).

This additional analysis is now added to the manuscript (please see figures 3 and 5, and methods and results sections for a detailed description).

Reference:

Mazaheri and Jensen. Posterior α activity is not phase-reset by visual stimuli. PNAS 2006; <https://doi.org/10.1073/pnas.0505785103>

2) Related to this, the authors argue that there is significant phase-locking of responses across subjects in the FEF-TMS condition (Figure 3B). However, it seems that this comparison is done to baseline (i.e. zero phase consistency), when it should be compared against the control stimulation condition (Cz-TMS) to appropriately control for any general effects of TMS stimulation.

R: To test for phase-locking across subjects we used the Rayleigh test for non-uniformity of circular data. We are therefore checking for a phase reset in each condition separately. We found that when FEF is stimulated the phase of the cosine models explaining the data are non-uniformly distributed, but this is not the case when FEF stimulation is replaced by Cz stimulation.

The reviewer correctly points out that we should ideally also test whether the magnitude of the phase reset (non-uniformity) is significantly bigger following FEF activation than after Cz stimulation, as we hypothesise that in the control (Cz-TMS) condition there is no or less phase reset (more uniformly distributed phase data) compared to FEF-TMS. However, to our knowledge, there is no appropriate statistical test that would allow us to perform such a comparison of uniformity of phase metrics. The Watson-Williams test for between-group comparisons of circular data, which would be equivalent to a t-test comparing two datasets, is designed to assess whether the mean angles of two groups are identical or not, but does not compare uniformity of angles across groups, and hence is uninformative for our purpose. Alternatively, we could calculate the pairwise difference between the angles extracted for each condition (using the function `circ_dist` provided in the Circular Statistics toolbox) for 16 and 17Hz cosine models, for which the Rayleigh test was significant. We could then test whether this difference is different from 0 (using the function `circ_mtest`). However, this approach is not ideal, as if there is no systematic phase alignment in the control condition, then by subtracting it from the main condition, we add noise to our data. Unsurprisingly, when performing this analysis, we found no significant results for both frequencies.

In the absence of appropriate tests for comparing uniformity of phase metrics across conditions, we implemented another test for the presence of oscillatory activity (applicable to EEG signals) that we can directly compare between two conditions (FEF vs sham) (please see next point). We hope that with this addition and the addition of the non-oscillatory (control) models for comparison to the cosine model (see point 1 above), we have convinced the reviewer of the interpretability of our beta effects in an oscillatory framework and of our effects to result from FEF-activation.

3) Also related to comment #1, what did the raw evoked potential look like over occipital cortex (i.e. before the time-frequency decomposition in Figure 1). Did it look like a sustained oscillation? This is a particular concern given the long time windows used to estimate phase (500 ms), which could spread out an impulse response over a broader time window.

R: To test for the presence of oscillatory activity in the evoked responses, we applied the BOSC method (“Better OSCillation detection”) to the 4 occipital electrodes showing a significant beta phase-realignment in the ITCP analysis (O1, Oz, O2, Iz. See Figure 1B). For a detailed description of the method please see Whitten et al.(2011).

Briefly, BOSC is designed to take into account the functional form of the “background” activity, i.e. the non-rhythmic portion of the signal, to detect the segments that deviate significantly from the spectral characteristics of the background. True oscillatory episodes are defined according to a duration and a power threshold, and therefore any increase in spectral amplitude that is non-repeating over time is rejected. The minimum duration was set to 3 cycles, while the power threshold was estimated for each frequency and electrode. First, a wavelet analysis was performed on the entire epoch (-1.5 to 1.5 sec, where 0ms is the time at which TMS was delivered) covering frequencies from 8 to 24Hz in 17 log-spaced steps. The background spectrum was modelled as coloured noise and fit to the actual power with a linear regression in log-log units. The power threshold for each frequency was calculated as the 95th percentile of the theoretical χ^2 distribution of wavelet power values, assuming the mean background spectrum estimated by the linear regression would be the mean of the corresponding χ^2 distribution at a given frequency. We

then extracted the proportion of time during which oscillations at a given frequency were present (P-episodes) in the 300-ms time-window we found the TMS to induce the phase reset.

This calculation was run separately for the real and sham condition, which were then compared across the frequencies and channels of interest, using a cluster corrected two-sided t-test. The figure below shows the percentage of oscillatory episodes (P-episodes) for each frequency. The grey rectangles indicate significant differences between sham and real stimulation and show that real TMS evokes beta oscillations over the occipital electrodes.

As shown in the figure, the TMS-evoked potentials over the occipital electrodes was associated with a beta oscillation for at least 50% of the time. The P-episodes in this band was also significantly different from the sham condition.

This new analysis and results have now been added to the manuscript, and the Figure is presented as Supplemental Material (Supplemental Figure S1).

Reference

Whitten T, Hughes A, Dickson CT, Caplan JB. A better oscillation detection method robustly extracts EEG rhythms across brain state changes: the human alpha rhythm as a test case. *NeuroImage* 2011;

<http://doi.org/10.1016/j.neuroimage.2010.08.064>

4) The authors argue that causal drive of FEF leads to phase-resetting and that this modulates perception. However, the effect of attention on detection tasks is known to change both the perceptual sensitivity of subjects (d') as well as the threshold of detection (criterion). The current approach does not allow the authors to distinguish between these hypotheses (or if both occur). This could be addressed by adding a more complex behavioral task that would be sensitive to either change.

R: We agree with the reviewer that it would be of interest to address whether FEF activation changed d' or criterion. We intend to use more complex behavioral tasks in future studies.

However, two previous studies have investigated the effects of single-pulse FEF-TMS on visual detection task and used d' and criterion as outcome measures (Chane et al., 2012 and Quentin, et al., 2013). Both reported an increase in d' when TMS was applied 80ms before the target presentation, but no change in the criterion. Despite the differences in the design, their results seem to suggest that the fluctuation of motion discrimination accuracy and phosphene perception rate are likely due to a fluctuation in perceptual sensitivity. Nonetheless, future studies should further address this point and try to directly test whether beta activity is related to sensitivity or bias.

References

Chanes L, Chica AB, Quentin R, Valero-Cabré A. Manipulation of Pre-Target Activity on the Right Frontal Eye Field Enhances Conscious Visual Perception in Humans. PlosOne 2012; <https://doi.org/10.1371/journal.pone.0036232>

Quentin R, Chanes L, Migliaccio R, Valabrègue R, Valero-Cabré R. Fronto-tectal white matter connectivity mediates facilitatory effects of non-invasive neurostimulation on visual detection. NeuroImage 2013; <http://dx.doi.org/10.1016/j.neuroimage.2013.05.083>

5) The authors do a thorough job of citing the relevant research around oscillations and attention, with a particular focus on the systems neuroscience literature. While I think this is appropriate, I felt that some of the existing literature on the effects of TMS (either single-pulse or repetitive) on oscillations in visual cortex was a bit light. Perhaps this stood out to me as the authors play a leading role in this literature. In particular, I thought the existing literature on alpha (~10 Hz) rhythms was missing. Furthermore, previous single pulse studies in FEF have shown an increase in perception time-locked to FEF stimulation (e.g. Grosbras and Paus, EJM 2003; also at ~10 Hz).

R: Thanks for pointing this out. The incomplete coverage was mainly due to the journal format which limits the number of words for introduction and discussion, but also the total number of references. To accommodate this point, we have now included more papers from the literature on FEF-TMS effects on visual cortex activity in the discussion section, also following reviewer2's suggestion. In particular, we now discuss papers that have described effects on occipital alpha (please see our response to comment 6).

Regarding the existing literature on single-pulse TMS applied to FEF, several studies provide evidence that the activation of this area improves behavioural performance in visual detection tasks (including the paper mentioned by the reviewer, Grosbras and Paus 2003, but also Quentin et al., 2013; Chanes et al., 2012) and increases the excitability of the visual cortex as shown by a decrease in phosphene threshold over V5 (Silvanto et al., 2006). The use by these studies of only a few intervals between FEF stimulation and target presentation (or phosphene testing) makes it difficult to directly compare our results with these studies. For example, in Grosbras 2003 only one interval was tested, i.e. FEF was stimulated 40 ms before the target presentation. Additional intervals (100, 130, 240ms before the target onset) were only tested in a small sample (3 participants). Furthermore, the sampling is very low (and not evenly spaced) and even though the figure we believe the reviewer is referring to (Figure 3) seems to suggest an alpha periodicity in d' , it is possible that faster frequencies are not visible because they would require a higher number of sampled data points (-100 and -40 might be two peaks of a beta oscillation at 17Hz). However, we do not see our results as refuting what has been previously reported, but rather a development.

We built upon the knowledge that FEF TMS is known to generate changes in the visual areas and tested a longer time window (and more intervals) to show that its influence is not discrete and limited to one interval, but rather follows an oscillation thus causing increase and decrease in visual performance and visual cortex excitability.

6) Related to this, a lot of previous work has shown TMS of FEF modulates alpha-band rhythms in occipital cortex. So, it is a bit surprising that the current experiment does not show any such modulation. Do the authors have an explanation for this discrepancy? I feel this should at least be discussed.

R: To the best of our knowledge, only a few studies have applied TMS over the FEF and recorded electrophysiological signals. Marshall et al, 2015 and Sauseng et al, 2011 have both applied an off-line inhibitory TMS protocol to investigate the consequences of FEF decreased activation on behavioural and MEG/EEG measures. While Marshall et al. chose cTBS and MEG recordings, Sauseng et al. applied 1hz TMS for 15 minutes and recorded EEG afterwards. Regardless of the TMS protocol, both studies found a reduction of the anticipatory alpha modulation over the occipital electrodes, during a visuospatial cued attention task. Finally, Capotosto et al. (2009) applied an on-line TMS protocol with short TMS train at 20Hz over the right FEF and investigated behavioural and EEG modulations after each train. rTMS was applied to interfere with the allocation of covert attention and during the cue-target interval. The results again showed a suppression of the well-known alpha modulation over the occipital electrodes. However, in all 3 studies, the authors decided to limit the analysis of the MEG/EEG data to a predefined frequency band, namely alpha (8-12 in Marshall et al., 10-12 in Sauseng et al., and low-high alpha individually adjusted around the individual alpha frequency in Capotosto et al.). While this selection was justified by the study hypothesis, it does not allow for a direct comparison with our EEG results, where no selection was applied to the frequency analysis. Nevertheless, even if there is no information regarding the beta frequency, these 3 studies clearly show that alpha is modulated when TMS is used to interfere or modulate FEF activity. One important difference with our study is the task that participants were asked to perform. In the previous studies, a cue informed the participants of the to-be-attended and, hence, to-be-ignored spatial location. In addition, an important task component in all these studies was sustained attention deployment. In our study, some of these elements are missing (no information to be filtered out, no sustained task component), which might explain the lack of modulation in the alpha-band. This would be in line with previous findings indicating that beta activity could be related to the processing of relevant stimuli (i.e. target)(Gross,2004; Siegel, 2008), while alpha oscillations might be involved when the task at hand requires the inhibition of competing information (distracters or locations) (Siegel, 2012). This is now better discussed in the revised manuscript.

References:

Marshall TR, O'Shea J, Jense O, Bergmann TO. Frontal eye fields control attentional modulation of alpha and gamma oscillations in contralateral occipitoparietal cortex. *J Neurosci* 2015; <http://doi.org/10.1523/jneurosci.3116-14.2015>

Sauseng P, Feldheim JF, Freunberger R, Hummel FC. Right prefrontal TMS disrupts interregional anticipatory EEG alpha activity during shifting of visuospatial attention. *Front. Psychol* 2011; <https://doi.org/10.3389/fpsyg.2011.00241>

Capotosto P, Babiloni C, Romani GL, Corbetta M. Frontoparietal cortex controls spatial attention through modulation of anticipatory alpha rhythms. *J Neurosci* 2009; <https://doi.org/10.1523/JNEUROSCI.0539-09.2009>

Gross J, Schmitz F, Schnitzler I, Kessler K, Shapiro K, Hommel B, Schnitzler A. Modulation of long-range neural synchrony reflects temporal limitations of visual attention in humans. *PNAS* 2004; <https://doi.org/10.1073/pnas.0404944101>

Siegel M, Donner TH, Oostenveld R, Fries P, Engel AK. Neuronal synchronization along the dorsal visual pathway reflects the focus of spatial attention. *Neuron* 2008; <https://doi.org/10.1016/j.neuron.2008.09.010>

Siegel M, Donner TH, Engel AK. Spectral fingerprints of large-scale neuronal interactions. *Nature reviews. Neuroscience* 2012; <https://doi.org/10.1038/nrn3137>

7) Previous psychophysical work has shown perceptual modulation from stimulus onset is at much lower frequencies (e.g. \sim theta in Landau et al). From Figure 3 and 4 it seems as if the authors focused on higher frequency rhythms (> 7 Hz). Did the authors test lower frequencies? In particular, there appears to be some late phase alignment at \sim 4-5Hz in Figure 1B and perhaps some modulation of phosphene detection from V1-TMS at \sim 5 Hz in Figure 4B (oddly maybe greater in the control condition?).

R: Many thanks for this important point. We did not find any significant phase-reset over occipital cortex in these lower (theta) frequency bands after FEF-activation (see EEG data, experiment 1). As a result, we decided to focus on the alpha and beta bands in the follow-up behavioural and phosphene experiments. Accordingly, we have optimized the design of the corresponding experiments (experiment 2-4) to resolve signals in the alpha and beta frequency bands. More specifically, we choose a 200ms window to cover at least 1.5 cycles of low alpha activity (\sim 7Hz) and a high sampling rate (experiment 2: 11.78ms steps/83Hz, experiment 3-4: 15ms steps, 67Hz) to cover frequencies well into the beta band. Our decision to inform the test window and intervals (and therefore the tested frequency) in experiments 2-4 by the EEG results is now better described in the methods section.

Due to these choices, we cannot confidently resolve frequencies <alpha by additional analyses. However, given the absence of any theta-phase reset in experiment 1, we are confident that this frequency is not evoked over occipital areas by FEF-TMS. We discuss now more carefully in the corresponding discussion section the differences in findings of our and previous studies (beta vs theta oscillations), alongside possible explanations (also to respond to point 8 below).

8) Related to comment #7, the authors note previous work showing phase alignment of behavioral responses to sudden onset visual stimuli in the discussion. However, they gloss over the fact that this occurs at a much different frequency. I think there are very reasonable explanations for this discrepancy (e.g. nesting of oscillations) but this should at least be noted and discussed.

R: We agree with the reviewer, and we have now included a section in our discussion, where we acknowledge and discuss the difference with previous studies.

9) The authors conclude that the period of phase alignment in EEG (Figure 1) coincides with the effects on phosphene perception (Figures 2 and 3). However, this isn't explicitly shown – what is the phase of the beta oscillation seen in Figure 1? Does it generally coincide with the expected phase given the periodicity in excitability seen with TMS of visual cortex?

R: In the original manuscript, we tested two independent samples, one for the EEG and one for the phosphene detection experiment, therefore we did not explicitly look at the phase values in the EEG signal. In the new version of the manuscript, the same participants took part in the EEG and motion detection task which allowed us to address this point. We have added additional figures (supplementary material) showing at the same time the phase distribution of the EEG signal (recorded during the real TMS condition) and the behavioural fluctuations, for those frequencies that showed a significant phase alignment in the behavioural performance (14-17Hz). As can be seen in the corresponding Figure (supplemental Fig 2), their respective phases tend to coincide and there was also a correlation between them for signals at 15Hz.

10) Do the authors control for the number of comparisons across frequency (e.g. Figure 3A)?

R: For the comparison between conditions, now figure 4C and 6C, R-squared labels from the two conditions were permuted and the difference of the mean calculated at each iteration. The null and real data were then compared and real data considered being significant if above the 97.5th percentile.

11) R-squared is bounded and therefore highly nonlinear, making it difficult to interpret differences, such as the quality of fit of cosine model for FEF-TMS and Cz-TMS (e.g. Figure 3A). The authors do an appropriate permutation test that should partially control for this issue, but did the authors also test other statistics of model fit (e.g. log-likelihood or percent explained variance)?

R: We only analysed the R-squared values. As suggested by the reviewer, we have compared the two conditions (now shown in figure 4C) using the log-likelihood calculated for each participant and condition as: $\text{Log-likelihood} = \log(\text{Root Mean Square error} / \text{Number of observations})$. Statistical differences were again tested by permuting the labels from the two conditions and calculating the difference in log-likelihood at each iteration. The null and real data were then compared and real data considered significant if above the 97.5th percentile. The figure below shows the results with significant differences between conditions highlighted in blue. As the results are very similar to what we originally reported in the manuscript, we would like to keep the original analysis and figure.

12) Related to this, what were the raw R-squared for the different frequency cosine models in the attention and control conditions? Was there a peak at the 16-17 Hz in the attention condition?

R: In the revised manuscript, we present a figure depicting the spectral content of the phosphene perception rate in the attention condition (Figure 5), clearly showing a peak at 16-17Hz.

REVIEWER COMMENTS

Reviewer #1 (Remarks to the Author):

I commend the authors for their extensive revision of the manuscript, including several new control experiments. In my view, all issues that I had raised on the original version of this work have been addressed satisfactorily.

One minor issue: In figure 6B, the inset shows erroneously a Cz-V5 placement of the two TMS coils, rather than a Cz-V1 placement.

Reviewer #2 (Remarks to the Author):

I want to congratulate the authors on the very extensive work done in the paper. The increased sample size in experiment 1 resolved my main original concern. The inclusion of a second experiment strengthened their findings even more. I have no further suggestions, and believe that this paper will be a great addition to the field.

Reviewer #3 (Remarks to the Author):

Top-down signals from the Frontal Eye Fields modulate visual cortex excitability by phase-realignment of oscillatory beta activity

Domenica Veniero, Joachim Gross, Stephanie Morand, Felix Duecker, Alexander Sack, and Gregor Thut

This is a re-review of this manuscript.

In brief, the authors investigate whether FEF stimulation resets the phase of top-down beta oscillations. They measure the impact of FEF TMS on both motion discrimination and phosphene induction. Both results show an effect at beta oscillations, around ~16-17 Hz.

As in the first round of reviews, I think this is an interesting paper that makes an important contribution to the literature. The addition of the motion discrimination task is valuable, demonstrating the behavioral impact of stimulation. In general, I think the revised manuscript addressed most of my previous concerns. However, a few more points of clarification remain.

1) Previously, I asked the authors to compare the oscillations model to other models. They did so by testing two more models of the evoked response as 1) an exponential decay or 2) an exponentially decay sinusoid. I thank the reviewers for following this suggestion and I think the new results provide support for the claim of oscillations in perception. However, it isn't entirely clear from the methods/results how these fits were done. The fit of the sinusoid was detrended by estimating and removing the linear trend from performance; was the same detrending applied before fitting the exponential and decaying sinusoid? If so, it isn't surprising that the exponential didn't fit as the detrended values will be mean zero. Instead, I would compare the exponential fit to the raw data to the current analysis.

2) Related to comment #1, if I'm reading the methods correctly, the decaying-sinusoid used a fixed tau of 0.01 milliseconds – why? Doesn't this cause the function to go to zero almost immediately? Regardless, why not fit this parameter?

3) Related to comment #1, the authors compare the r-squared values to assess the goodness of fit of the different models. This doesn't appear to have been adjusted for the number of

parameters in each model. I would suggest the authors either use a model comparison statistic (AIC/BIC), use adjusted r-squared, or cross-validation to make sure that the number of parameters of the model are accounted for when comparing the models. Note: the number of free parameters should include the detrending.

4) In my previous comment (#10), I asked whether the authors correct for multiple comparisons across the different frequencies when testing their model fits. Their answer seems to restate their statistical testing procedure but does not mention any correction for multiple comparisons. Can they please clarify in the text whether these are corrected values?

5) In my previous comment #2, I noted that the authors should directly compare the level of phase-locking during FEF stimulation compared to control (Cz) stimulation. In their response, they noted that this is difficult. I understand the difficulty in finding a parametric test to directly test the degree of phase-locking, but I was imagining an analysis that involved permuting across conditions. The hypothesis is the phase-locking effect was different in the FEF and Cz condition. If so, then the difference in uniformity across conditions (e.g., measured as the difference in average vector length) should be larger in the observed data than in permuted data where the results are shuffled across the FEF and Cz conditions (thus, breaking the correlation between stimulation condition and phase-locking).

REVIEWER COMMENTS

Reviewer #1 (Remarks to the Author):

I commend the authors for their extensive revision of the manuscript, including several new control experiments. In my view, all issues that I had raised on the original version of this work have been addressed satisfactorily.

One minor issue: In figure 6B, the inset shows erroneously a Cz-V5 placement of the two TMS coils, rather than a Cz-V1 placement.

R: We thank the reviewer for the positive comments and for spotting the mistake in Figure 6B, which has been amended.

Reviewer #2 (Remarks to the Author):

I want to congratulate the authors on the very extensive work done in the paper. The increased sample size in experiment 1 resolved my main original concern. The inclusion of a second experiment strengthened their findings even more. I have no further suggestions, and believe that this paper will be a great addition to the field.

R: We thank the reviewer for the positive comments. We are delighted to know he/she finds our paper interesting.

Reviewer #3 (Remarks to the Author):

Top-down signals from the Frontal Eye Fields modulate visual cortex excitability by phase-realignment of oscillatory beta activity

Domenica Veniero, Joachim Gross, Stephanie Morand, Felix Duecker, Alexander Sack, and Gregor Thut

This is a re-review of this manuscript.

In brief, the authors investigate whether FEF stimulation resets the phase of top-down beta oscillations. They measure the impact of FEF TMS on both motion discrimination and phosphene induction. Both results show an effect at beta oscillations, around ~16-

As in the first round of reviews, I think this is an interesting paper that makes an important contribution to the literature. The addition of the motion discrimination task is valuable, demonstrating the behavioral impact of stimulation. In general, I think the revised manuscript addressed most of my previous concerns. However, a few more points of clarification remain.

R: Many thanks for giving us the opportunity to provide clarifications. These are detailed point-by-point and integrated in the manuscript as described below. The changes are highlighted in the new text in red font.

1) Previously, I asked the authors to compare the oscillations model to other models. They did so by testing two more models of the evoked response as 1) an exponential decay or 2) an exponentially decay sinusoid. I thank the reviewers for following this suggestion and I think the new results provide support for the claim of oscillations in perception. However, it isn't entirely clear from the methods/results how these fits were done. The fit of the sinusoid was detrended by estimating and removing the linear trend from performance; was the same detrending applied before fitting the exponential and decaying sinusoid? If so, it isn't surprising that the exponential didn't fit as the detrended values will be mean zero. Instead, I would compare the exponential fit to the raw data to the current analysis.

R: Apologies if the description of the fitting was not clear. And many thanks for raising the point of detrending the behavioural data before fitting an exponential to it, which made us reconsider the exponential decay fit (not the exponential decay sinusoid), as explained below.

In response to the very well taken point by this reviewer (from the former round of comments), that it is crucial to demonstrate that our results cannot be explained by an ERP evoked in the visual cortex due to FEF-stimulation, we ran a number of additional analyses for our previous revisions. In the EEG data, we have looked for a sustained oscillatory activity lasting at least 3 cycles (using BOSC analysis) over the occipital electrodes after FEF-stimulation and found that this is present in the beta band. To demonstrate that the same is true for the behavioural data, we compared an oscillatory model of motion discrimination accuracy and phosphene perception rate to two non-oscillatory models, namely an ERP model (exponentially decaying sinusoid, as implemented by Mazaheri and Jensen, PNAS, 2006) and a pure exponential model. For both the oscillatory model and the ERP model (in which deflections are expected to fluctuate at relatively fast theta-to-beta frequencies), we had to detrend our data prior to performing any fit. The linear detrend eliminates slow (low frequency) behavioural fluctuation that may be present in the behavioural data and would conceal the faster

oscillatory or ERP changes, hence affecting oscillatory or ERP model fits. As pointed out by the reviewer, we therefore detrended the data prior to performing any fit.

We agree that detrending before fitting a pure exponential is not appropriate, but this well taken point also made us re-think whether fitting a pure exponential to our data made sense in the first place. Our aim is to demonstrate that there is some correspondence between the EEG and behavioural data in terms of oscillatory responses that are elicited by FEF-activation (here in the beta bands) and, in analogy to the EEG data, we must exclude that an ERP can explain the behavioural data. Here, the comparison between the oscillatory and the ERP model of our behavioural data is crucial, as it complements and completes our EEG analysis. In hindsight, however, the third fit of a pure exponential model does not seem well aligned with our goals. This model is sensitive to slow fluctuations, such as TMS-induced changes in alertness that can occur e.g., due to the alerting TMS pulse which may lead to better performance closer to TMS delivery and a slow performance decay after. However, this is not what we observed in the EEG data (no indication of TMS-locked slow drifts that would require analysis of slow performance changes over time) but is what we wanted to exclude from our data in the first place (by detrending to pick up faster oscillatory activity riding on the slow changes).

In brief, in hindsight, we realize that the pure exponential fit does not add anything to our arguments at best (neither supports nor weakens our results/conclusions) or is confusing at worst, and hope the reviewer agrees. This is why we have removed it from our analysis, while keeping the ERP model for comparison with the cosine model.

By extension, we kept detrending before fitting (but tried to be clearer why this was implemented in the methods/results section). Please see directly in the manuscript (changes highlighted in red font).

2) Related to comment #1, if I'm reading the methods correctly, the decaying-sinusoid used a fixed tau of 0.01 milliseconds – why? Doesn't this cause the function to go to zero almost immediately? Regardless, why not fit this parameter?

R: Many thanks for spotting this mistake. We erroneously wrote 0.01 *milliseconds*, when the correct time unit should have been *seconds* (i.e. 0.01s). Hence, the decay factor in our model was 10 milliseconds. This mistake is now corrected in the new version.

In addition, we are now fitting the decay factor as suggested, while setting the upper bound to 10ms. It is worth noting that this parameter will determine how many peaks and troughs the model will have, so this choice ensured that the non-oscillatory ERP model did not overlap with the oscillatory model; meaning that the non-oscillatory ERP model will be limited in peaks and troughs, while the oscillatory model will not be limited in peaks and troughs. For an example of best fits of the oscillatory and non-oscillatory

ERP models in a representative participant, please see Figure 1 below (which is now also integrated in the new manuscript).

We chose 10ms as the upper limit for two reasons: One guided by the time-scale of our behavioural data and the other based on empirical grounds. In particular, 10ms is in the same order of magnitude as our timescale, given that behavioural data points were sampled every 15ms. Moreover, because the aim of this fitting was to test if the results can be explained by an evoked potential, we also chose the value to lead to an ERP shape that would resemble what has been used in the relevant paper by Mazaheri and Jensen (Mazaheri A, Jensen O. 2006. Posterior alpha activity is not phase-reset by visual stimuli. Proc Natl Acad Sci U S A 103, 2948-2952). See figure 6B of that paper, copied below as Figure 2 for comparison.

Figure 1. Examples of best oscillatory and non-oscillatory (ERP) model fits in a representative participant (participant 1). The blue line represents the best fitting ERP model for this participant, the red line the best fitting cosine model (lowest AIC values).

For comparison:

Figure 2 Mazaheri and Jensen's ERP model.

3) Related to comment #1, the authors compare the r-squared values to assess the goodness of fit of the different models. This doesn't appear to have been adjusted for the number of parameters in each model. I would suggest the authors either use a model comparison statistic (AIC/BIC), use adjusted r-squared, or cross-validation to make sure that the number of parameters of the model are accounted for when comparing the models. Note: the number of free parameters should include the detrending.

R: We agree with the reviewer that the best way to compare models must consider the number of fitted parameter. This is now implemented in our new version.

We have decided to use Akaike's Information Criterion (AIC) to compare the 2 models. For the motion discrimination task (experiment 2), the number of observations is equal 18 (18 time points in each window) for all models, while the number of parameters is equal 6 for the ERP model (detrend, decay factor (τ), T_0 , frequency, phase, amplitude) and equal 4 for the cosine model (detrend, frequency, phase, amplitude). Note that we selected the best model for each participant amongst all possible frequencies in the significant range (15-23Hz for window1 and 16-24 for window2). The best model was defined as the one yielding the lowest AIC. For both windows, the oscillatory model is significantly better than the ERP model ($p < 0.05$).

The results of this comparison are shown below.

For phosphene detection rate (experiment 3), we ran the same analysis, but with a number of observations equal to 14 (14 time points in window 1). As for motion discrimination performance, the oscillatory model outperformed the ERP model in terms of fit to the data. The results are shown below.

We have changed the relevant manuscript sections (methods and results) and figures 3 and 5 to accommodate these changes.

4) In my previous comment (#10), I asked whether the authors correct for multiple comparisons across the different frequencies when testing their model fits. Their answer seems to restate their statistical testing procedure but does not mention any correction for multiple comparisons. Can they please clarify in the text whether these are corrected values?

R: In the analysis that we report in Figure 4c and 6c, we compared the r-squared values obtained from the cosine fit for frequencies ranging from 7 to 25Hz between the 2 conditions, i.e., FEF-V5 vs Cz-V5 or FEF-V1 vs Cz-V1 stimulation. We used a permutation approach where the real difference between conditions is compared to a null distribution obtained by shuffling the labels of the 2 conditions. As pointed out by the reviewer, we did so for each frequency. We then considered as significant the cosine models for which the value of the real difference between conditions exceeded the 97.5th percentile of the null distribution. As this procedure does not provide a p-value, we did not correct this value for multiple comparisons.

Following the reviewer comment, we have now added a second analysis that tests for differences between conditions taking the overall data into account, before testing simple effects at multiple frequencies. We performed two repeated-measures ANOVAs with the factors TMS condition and frequency, one for experiment 3 and one for experiment 4. The results of the ANOVAs and follow-up tests are now reported in the revisions in support of the permutation results (we also clearly state that follow-up t-tests are uncorrected for multiple tests). The new results are detailed below:

Experiment 3: V5 excitability changes.

First time-window (Figure 4C). We performed a repeated measures ANOVA with factors *Condition* (2 levels: FEF-V5, Cz-V5) and *Frequency* (19 levels: cosine frequencies from 7 to 25 Hz). The results indicate a significant interaction *Condition*Frequency* ($F_{(18,90)}=2.737$, $p=0.001$), but no main effects of *Condition* ($F_{(1,5)}=0.375$; $p=0.576$) or *Frequency* ($F_{(18,90)}=0.483$, $p=0.959$).

To follow up on the significant interaction, we performed two-tailed t-tests to compare the r-squared value of each frequency model between conditions (uncorrected for multiple tests). For the models with frequency between 7-15Hz this difference was not statistically significant (p-values: 0.4-0.8). FEF-V5 and Cz-V5 were significantly different for the 16Hz ($t=3.04$, $p=0.029$), 17Hz ($t=3.07$, $p=0.028$) and 18Hz model ($t=2.86$, $p=0.035$). For higher frequencies between 19-25Hz, the two conditions again did not differ (p-values: 0.052-0.8). The lowest p-value=0.052 was obtained when comparing the 19Hz models.

Second time-window (Figure S3b). A repeated measure ANOVA with factors *Condition* and *Frequency* was also performed for the second time-window. There was no significant *Condition*Frequency* interaction ($F_{(18,90)} = 0.952$, $p=0.521$), and no main effects of *Condition* ($F_{(1,5)}=0.152$, $p=0.712$) or *Frequency* ($F_{(18,90)}= 0.574$, $p=0.910$).

Experiment 4: V1 excitability changes.

First time-window (Figure 6C). As for experiment 3, we performed a repeated measures ANOVA with factors *Condition* (2 levels: FEF-V1, Cz-V1) and *Frequency* (19 levels: cosine frequencies from 7 to 25 Hz). In this case, we found no significant *Condition*Frequency* interaction ($F_{(18,108)}=0.24$, $p=0.999$). We found no effect of *Condition* ($F_{(1,6)}=0.084$, $p=0.782$), but a main effect of *Frequency* ($F_{(18,108)}= 4.961$, $p=0.00$) explained by a difference between some of the models regardless of the conditions, in particular 8 and 9Hz having lower r-squared values compared to 7Hz and 16-25Hz. As this is not of interest, we are not reporting any detailed follow-up analysis of these results. It is worth noting that from Figure 6C it is clear that the two TMS conditions (FEF-V1 and Cz-V1 control) show an almost identical pattern, and their difference is close to 0 across all frequency models.

Second time-window (Figure S3d). We did run the same ANOVA on the second time window (figure S3d) and found again no interaction *Condition*Frequency* ($F_{(18,108)}= 0.373$; $p=0.990$).

In the new manuscript, we have added this information in summarized form to the results and methods section in support of the permutation analysis.

5) In my previous comment #2, I noted that the authors should directly compare the level of phase-locking during FEF stimulation compared to control (Cz) stimulation. In their response, they noted that this is difficult. I understand the difficulty in finding a parametric test to directly test the degree of phase-locking, but I was imagining an analysis that involved permuting across conditions. The hypothesis is the phase-locking effect was different in the FEF and Cz condition. If so, then the difference in uniformity across conditions (e.g., measured as the difference in average vector length) should be larger in the observed data than in permuted data where the results are shuffled across the FEF and Cz conditions (thus, breaking the correlation between stimulation condition and phase-locking).

R: We have tried to analyse the data according to the reviewer's suggestion. We found that the average vector length is not different in the 2 conditions. While we understand the reviewer's comment, we would like to note 2 points regarding the phase analysis.

1- Summarizing the phosphene data, we have demonstrated that there is a periodicity in the phosphene perception rate at beta frequency. We also showed that this is not explained by TMS-unspecific effects because the main and control conditions are significantly different for the beta frequency. These results alone support the hypothesis

of a phase reset, otherwise no periodicity would be seen in behaviour. The fact that the phase is consistent across participants (as shown by the phase locking analysis) is hence not essential to support the hypothesis of a beta phase reset as a top-down mechanism, but it is a nice addition to the results.

2- The second point is that the two conditions clearly differ in the beta band (see e.g., figure 4 and related analysis) with beta oscillations significantly explaining phosphene fluctuations in the main but not the control condition. This is the reason why we did not examine the phase in the control condition.

In brief, our rationale was to proceed stepwise and only considered the oscillatory activity that was significantly different between conditions, then testing if this activity was explaining the single conditions and only if these 2 criteria were satisfied, we proceeded with the phase consistency analysis.

Overall, we strongly feel that the phase consistency analysis adds to the present data and would like to keep it in, even if not explored at all levels of the data (for the above stated reasons) and not ultimately essential to make our points. We hope the reviewer agrees.

REVIEWERS' COMMENTS

Reviewer #3 (Remarks to the Author):

The authors have addressed all of my previous concerns. The addition of AIC makes the model comparison stronger. However, it is not statistically valid to directly test AIC distributions; they themselves are measures of statistical difference (e.g., you wouldn't test if p-values were significantly different). The difference in AIC values shown (~ 10) are typically taken as fairly strong evidence for the better-performing model.

REVIEWER COMMENTS

Reviewer #3 (Remarks to the Author):

The authors have addressed all of my previous concerns. The addition of AIC makes the model comparison stronger. However, it is not statistically valid to directly test AIC distributions; they themselves are measures of statistical difference (e.g., you wouldn't test if p-values were significantly different). The difference in AIC values shown (~10) are typically taken as fairly strong evidence for the better-performing model.

R: We would like to thank the reviewer for pointing this out. We have now removed the statistics we had originally performed on AIC scores.

The results are now reported in terms of difference in AIC scores between the two models as follow. For experiment 2, the difference between ERP AIC and cosine AIC is $\Delta=10$ for window1 and $\Delta= 11.6$ for window 2. For experiment 3, $\Delta=9.2$. As suggested by the reviewer and according to Burnham & Anderson (2002), a difference between models that is ~10 is considered strong evidence in itself that the model with lower AIC values is better describing the data and therefore does not require any follow-up analysis.

Burnham, K.P. & Anderson, D.R. 2002. Model Selection and Multimodel Inference: A Practical Information-Theoretic Approach, 2nd edn. Springer, Berlin.